# Towards Efficient LLM Grounding for Embodied Multi-Agent Collaboration

## Abstract

Grounding the reasoning ability of large language models (LLMs) for embodied tasks is challenging due to the complexity of the physical world. Especially, LLM planning for multi-agent collaboration requires communication of agents or credit assignment as the feedback to re-adjust the proposed plans and achieve effective coordination. However, existing methods that overly rely on physical verification or self-reflection suffer from excessive and inefficient querying of LLMs. In this paper, we propose a novel framework for multi-agent collaboration that introduces Reinforced Advantage feedback (ReAd) for efficient self-refinement of plans. Specifically, we perform critic regression to learn a sequential advantage function from LLM-planned data, and then treat the LLM planner as an optimizer to generate actions that maximize the advantage function. It endows the LLM with the foresight to discern whether the action contributes to accomplishing the final task. We provide theoretical analysis by extending advantage-weighted regression in reinforcement learning to multi-agent systems. Experiments on Overcooked-AI and a difficult variant of RoCoBench show that ReAd surpasses baselines in success rate, and also significantly decreases the interaction steps of agents and query rounds of LLMs, demonstrating its high efficiency for grounding LLMs. More results are given at `https://read-llm.github.io/`.

## 1 Introduction

Large Language Models (LLMs) have exhibited remarkable capabilities across various domains, including long-text understanding, reasoning, and text generation (Devlin et al., 2019; Radford et al., 2019; Brown et al., 2020; Raffel et al., 2020). Benefiting from large-scale text corpora mined from the web, LLMs can absorb and capture vast quantities of knowledge about the world for decision-making. Recent research has shown that LLMs can interactively make decisions through zero-shot or few-shot example prompting to solve embodied tasks (Firoozi et al., 2023) via chain-of-thought (CoT) (Wei et al., 2022) or tree-of-thought (Yao et al., 2023a) planning. However, LLMs perform planning only using their internal knowledge, which is often not grounded in the physical world due to the lack of task-specific knowledge of complex embodied agents. Such a problem can lead to fact hallucination and nonsensical instruction interpretation issues in reasoning (Ahn et al., 2022). To prevent LLMs from outputting infeasible plans in embodied tasks, existing methods mostly design a closed-loop framework for the interaction process with feedback. Specifically, one line of research adopts *self-reflection* by performing self-evaluation by LLMs to improve the plan generation of LLM planner (Shinn et al., 2023; Yao et al., 2023b; Hao et al., 2023; Liu et al., 2023b); and the other works perform *physical verification* by using feedback of the external environment to dynamically replan depending on unexpected feedback (Huang et al., 2022b; Song et al., 2023a). Nevertheless, these feedback is often sparse or designed heuristically, a more principled feedback mechanism for LLM-based embodied task planning is still lacking.

Considering more challenging planning problems in multi-agent settings, an LLM-based agent needs to cooperate with other agents through communication and negotiation, which causes more difficulties in effective feedback. Specifically, it is hard for both self-reflection and physical verification to evaluate the effects of individual action in a team outcome of multi-agents. Consequently, the feedback mechanisms suffer from either excessive queries of LLMs or frequent interactions with the physical environment. For instance, RoCo (Mandi et al., 2023) introduces physical verification as feedback to refine the LLM-generated actions in multi-agent cooperative settings, but faces the

(a) The task snapshot     (b) Multi-Agent negotiation with env. feedback (RoCo)     (c) Multi-Agent negotiation with Adv. feedback (Ours)

Figure 1: An illustration of the negotiation process of RoCo and our method. RoCo interacts with the environment for each plan and takes the environment's feedback as prompts. In contrast, our method takes the advantage function (Adv.) evaluated by a critic as feedback, and revises the plan if the advantage value is lower than the threshold, which significantly reduces the interaction rounds to the environment.

difficulty of poor efficiency. As we illustrated in Figure 1, RoCo requires excessive interaction to obtain physical feedback and queries to LLMs to get feasible joint-action plans, which can be heavily inefficient for embodied tasks. In contrast, various methods in Multi-Agent Reinforcement Learning (MARL) (Zhang et al., 2021) have developed value or advantage decomposition theories for credit assignment of multiple agents (Rashid et al., 2020; Kuba et al., 2022a), which provide effective mechanisms to evaluate the contribution of individual actions in accomplishing final tasks and can generate actions for monotonic policy improvement (Kuba et al., 2022b). Inspired by these principles, we ask "*How to enhance the reasoning ability of LLMs for embodied multi-agent collaboration* with theoretical supports of MARL?". Our objective is to build an efficient feedback and refinement algorithm with utilizing multi-agent advantage functions, for multi-agent planning assisted by LLMs.

In this paper, we propose Reinforced Advantage (*ReAd*) as a closed-loop feedback for LLMs in multi-agent collaboration. We provide two optional LLM-generated plan refinement scheme, including Sequential Individual Plan Refinement with the local advantage (named *ReAd-S*) and Joint Plan Refinement with the joint advantage (named *ReAd-J*). Among them, (i) *ReAd-J* evaluates the advantage function of joint actions, which requires LLMs to generate the joint planning of all agents at once. In contrast, (ii) *ReAd-S* evaluates the local advantages of each agent's action by following the principle of multi-agent advantage decomposition (Kuba et al., 2022a) in MARL, which allows LLMs to generate actions for each agent sequentially. Both advantage functions are estimated by a critic network that regresses LLM-planned data. Based on the advantage function, an LLM planner is used as an optimizer by prompting to generate actions that maximize the advantage value. Otherwise, the LLM planner is required to re-plan if the advantage value is small. We provide a theoretical motivation for such a process by extending advantage-weighted regression (Peng et al., 2019) to multi-agent settings. In experiments, we extend RoCoBench (Mandi et al., 2023) to a difficult variant, which we term *DV-RoCoBench*. The results on *DV-RoCoBench* and *Overcooked-AI* show that *ReAd* significantly decreases the interaction and query rounds, and also surpasses baselines in success rate, highlighting its effectiveness for grounding LLMs in embodied multi-agent collaboration tasks.

## 2 PRELIMINARIES

We consider a Markov game, which is defined by a tuple $\langle \mathcal{N}, \mathcal{S}, \mathcal{A}, P, r, \gamma \rangle$, in which $\mathcal{N}$ denotes the set of agents, $\mathcal{S}$ denotes state space, $\mathcal{A} = \prod_{i=1}^{n} \mathcal{A}^i$ denotes the product of finite action spaces of all agents (i.e., joint action space), $P : \mathcal{S} \times \mathcal{A} \times \mathcal{S} \rightarrow [0, 1]$ denotes the transition probability function, $r : \mathcal{S} \times \mathcal{A} \rightarrow \mathbb{R}$ denotes the reward function, and $\gamma \in [0, 1)$ denotes the discount factor. In the Markov game, every agent at time step $t \in \mathbb{N}$ observes the state of environment $s_t \in \mathcal{S}$ and takes an action $a_t^i \in \mathcal{A}^i$ from its corresponding policy $\pi^i(\cdot|s_t)$, which together with other agents' actions forms a joint action $\boldsymbol{a}_t = (a_t^1, a_t^2, ..., a_t^n) \in \mathcal{A}$ drawn from the joint policy $\boldsymbol{\pi}(\cdot|s_t) = \prod_{i=1}^{n} \pi^i(\cdot|s_t)$. Then agents receive a shared reward $r_t = r(s_t, \boldsymbol{a}_t)$ and observe a new state $s_{t+1}$ with probability $P(s_{t+1}|s_t, \boldsymbol{a}_t)$. With the joint policy $\boldsymbol{\pi}$ and the transition probability function $P$, the state value function is defined as $V_{\boldsymbol{\pi}}(s) \triangleq \mathbb{E}_{s_{1:\infty} \sim P, \boldsymbol{a}_{0:\infty} \sim \boldsymbol{\pi}}[\sum_{i=0}^{\infty} \gamma^i r_i | s_0 = s]$. And the state-action value

function is defined as $Q_{\pi}(s, \boldsymbol{a}) \triangleq \mathbb{E}_{s_{1:\infty} \sim P, \boldsymbol{a}_{1:\infty} \sim \boldsymbol{\pi}}[\sum_{i=0}^{\infty} \gamma^i r_i | s_0 = s, \boldsymbol{a}_0 = \boldsymbol{a}]$. We aim at finding a joint policy to maximize the expected return $J(\boldsymbol{\pi}) \triangleq \mathbb{E}_{s_{0:\infty} \sim P, \boldsymbol{a}_{0:\infty} \sim \boldsymbol{\pi}} [\sum_{t=0}^{\infty} \gamma^t r_t]$. In the following, we consider the LLM planner as a special RL policy, which can be evaluated by a value function.

## 3 METHODOLOGY

We first give definitions and learning algorithms for the two kinds of advantage functions in §3.1. Then, we provide theoretical motivation for grounding LLMs by extending advantage-weighted regression in multi-agent settings in §3.2. Finally, we describe how to derive Reinforced Advantage (*ReAd*) feedback from the theoretical motivation and use an LLM planner as an optimizer and refine the plan in §3.3.

### 3.1 LEARNING OF ADVANTAGE FUNCTIONS

We first introduce the estimation of *joint* advantage function. Then the *local* advantage is obtained via advantage decomposition by following theories from MARL.

**Joint Advantage Function.** Based on joint value functions $Q_{\pi}(s, \boldsymbol{a})$ and $V_{\pi}(s)$, we define the *joint* advantage function as

$$A_{\pi}(s, \boldsymbol{a}) \triangleq Q_{\pi}(s, \boldsymbol{a}) - V_{\pi}(s),$$

which evaluates the advantage value of joint actions $\boldsymbol{a}_t = (a_t^1, a_t^2, ..., a_t^n)$ from all agents. $A_{\pi}(s, \boldsymbol{a})$ will be used for *ReAd-J* to evaluate the joint planning of all agents as feedback. Here, we assume the option of taking no actions is available to each agent, which is reasonable and common in embodied tasks. With this special action that we term WAIT, we can estimate the joint advantage using only $Q_{\pi}(s, \boldsymbol{a})$.

When taking WAIT action $a = w$, the agent will keep dormant at the current time step. The joint WAIT action is denoted as $\boldsymbol{w} = (w, w, ..., w)$. Choosing $\boldsymbol{w}$ at the current state $s$ signifies all agents take no actions, then the next state $s' = s$ and the agents receive shared reward $r(s, \boldsymbol{w}) = 0$ since $\boldsymbol{w}$ bring no changes to the environment. Further, we can derive the relationship between $Q_{\pi}(s, \boldsymbol{w})$ and $V_{\pi}(s)$, as

$$Q_{\pi}(s, \boldsymbol{w}) = \mathbb{E}_{s_{1:\infty} \sim P, \boldsymbol{a}_{1:\infty} \sim \boldsymbol{\pi}} \left[ \sum_{i=0}^{\infty} \gamma^i r_i \big| s_0 = s, \boldsymbol{a}_0 = \boldsymbol{w} \right]$$

$$= \gamma \mathbb{E}_{s_{2:\infty} \sim P, \boldsymbol{a}_{1:\infty} \sim \boldsymbol{\pi}} \left[ \sum_{i=0}^{\infty} \gamma^i r_{i+1} \big| s_1 = s \right] = \gamma V_{\pi}(s).$$

Therefore, the *joint* advantage function can be derived by using only the $Q_{\pi}$ function, as

$$A_{\pi}(s, \boldsymbol{a}) = Q_{\pi}(s, \boldsymbol{a}) - \frac{1}{\gamma} Q_{\pi}(s, \boldsymbol{w}). \tag{1}$$

**Local Advantage Function.** In cooperative multi-agent settings, we can further consider the contribution to performance in different subsets of agents' views. We adopt the standard definition in MARL to measure the local advantages.

**Definition 1.** (Kuba et al., 2022a) *Let $i_{1:m}$ denote an ordered subset $\{i_1, ..., i_m\}$ of $\mathcal{N}$, and let $-i_{1:m}$ refer to its complement. We mark $i_k$ when we refer to the $k^{th}$ agent in the ordered subset. Correspondingly, the multi-agent local state-action value function is defined as*

$$Q_{\pi}^{i_{1:m}}(s, \boldsymbol{a}^{i_{1:m}}) \triangleq \mathbb{E}_{a^{-i_{1:m}} \sim \pi^{-i_{1:m}}} \left[ Q_{\pi}(s, \boldsymbol{a}^{i_{1:m}}, \boldsymbol{a}^{-i_{1:m}}) \right] \tag{2}$$

*and for disjoint sets $j_{1:k}$ and $i_{1:m}$, the multi-agent local advantage function is*

$$A_{\pi}^{i_{1:m}}(s, \boldsymbol{a}^{j_{1:k}}, \boldsymbol{a}^{i_{1:m}}) \triangleq Q_{\pi}^{j_{1:k}, i_{1:m}}(s, \boldsymbol{a}^{j_{1:k}}, \boldsymbol{a}^{i_{1:m}}) - Q_{\pi}^{j_{1:k}}(s, \boldsymbol{a}^{j_{1:k}}) \tag{3}$$

**Monte Carlo Estimation.** Both Eqs. (1) and (3) can be estimated via the local value function $Q_{\pi}^{i_{1:u}}(s, \boldsymbol{a}^{i_{1:u}})$ with arbitrary action subset $\boldsymbol{a}^{i_{1:u}}$. More precisely, the local advantages can be estimated by changing $\boldsymbol{a}^{i_{1:u}}$ to disjoint action sets or subsets, and the joint advantages can be obtained by changing $\boldsymbol{a}^{i_{1:u}}$ to $\boldsymbol{a}^{1:n}$ that contains the joint actions or the joint WAIT action. In the following, we denote the underlying policy of the LLM planner as $\boldsymbol{\mu} = \boldsymbol{\pi}_{\text{llm}}(\boldsymbol{a}|s)$. To estimate $Q_{\boldsymbol{\mu}}^{i_{1:u}}$, we collect a dataset $\mathcal{D}$ by following the behavior policy $\boldsymbol{\mu}$, and further augment it with enhanced trajectories to

overcome the out-of-distribution (OOD) problem of action estimation (Levine et al., 2020). Then we estimate $Q_{\boldsymbol{\mu}}^{i_{1:u}}(s, \boldsymbol{a}^{i_{1:u}})$ via Monte Carlo estimation by following $\mathcal{R}_{s,\boldsymbol{a}^{i_{1:u}}} = \sum_{\boldsymbol{a}^{-i_{1:u}} \in \mathcal{D}} \sum_{t=0}^{T} \gamma^t r_t$, where the complement sets is sampled from the dataset. Then the value function is learned by a regression loss as

$$\mathbb{E}_{s,\boldsymbol{a}^{i_{1:u}} \sim \mathcal{D}} \left[ \left\| \mathcal{R}_{s,\boldsymbol{a}^{i_{1:u}}} - Q_{\boldsymbol{\mu}}^{i_{1:u}} \right\|^2 \right].$$

We refer to Alg. 1 in §C for the details. The setting of reward $r_t$ depends on the specific task, e.g., for sweeping cubes in Figure 1, $r_t = 1$ if a correct cube is swept and $r_t = 0$ otherwise. The details of data collection are given in §E.5.

**Advantage Decomposition.** Based on Eq. (2), we can express the state value function $V_{\boldsymbol{\pi}}(s)$ in a new form. Given the whole set of agents $\mathcal{N} = \{1, .., n\}$,

$$V_{\boldsymbol{\pi}}(s) = \mathbb{E}_{a^{1:n} \sim \boldsymbol{\pi}^{1:n}} \left[ Q_{\boldsymbol{\pi}}(s, \boldsymbol{a}^{1:n}) \right].$$

Based on Definition 1, we can introduce a pivotal lemma, which reveals that joint advantage function can be decomposed into the summation of local advantages of each agent.

**Lemma 1.** *(Multi-Agent Advantage Decomposition). In any cooperative Markov games, given a joint policy $\boldsymbol{\pi}$ and the whole set of agents $\mathcal{N} = \{1, .., n\}$, for any state $s$, and any ordered set $i_{1:n}$ of all agents, we have*

$$A_{\boldsymbol{\pi}}(s, \boldsymbol{a}) = \sum_{k=1}^{n} A_{\boldsymbol{\pi}}^{i_k}(s, \boldsymbol{a}^{i_{1:k-1}}, a^{i_k}), \tag{4}$$

*where $\boldsymbol{a} = (a^1, a^2, ..., a^n)$.*

The proof follows Kuba et al. (2022a) and is given in §A.1. Lemma 1 will be used for derivation in §3.2.

## 3.2 THEORETICAL MOTIVATION FOR GROUNDING LLM

In this section, we give a theoretical motivation that closely resembles advantage-weighted regression (Peng et al., 2019) in single-agent RL, while we extend it for multi-agents via advantage decomposition in Lemma 1. To achieve efficient LLM grounding, i.e., to obtain a superior policy to the LLM planner, one option is adopting LLM as a basic policy and searching for a stronger policy than it. Therefore, we derive our objective as an approximate optimization of a constrained policy search problem. Specifically, we denote the policy of LLM planners as $\boldsymbol{\mu} = \boldsymbol{\pi}_{\text{llm}}(\boldsymbol{a}|s)$, and our goal is to find a policy $\boldsymbol{\pi}$ that maximizes the expected improvement $\eta(\boldsymbol{\pi}) = J(\boldsymbol{\pi}) - J(\boldsymbol{\mu})$ over the basic policy $\boldsymbol{\mu}$. Following the performance difference lemma (Kakade & Langford, 2002; Schulman et al., 2015), we show the expected improvement $\eta(\boldsymbol{\pi})$ can be expressed in terms of the advantage over $\boldsymbol{\mu}(\boldsymbol{a}|s)$, as

$$\eta(\boldsymbol{\pi}) = \mathbb{E}_{s \sim \rho_{\boldsymbol{\pi}}(s), \boldsymbol{a} \sim \boldsymbol{\pi}(\boldsymbol{a}|s)} \left[ A_{\boldsymbol{\mu}}(s, \boldsymbol{a}) \right], \tag{5}$$

where $\rho_{\boldsymbol{\pi}}(s) = \sum_{i=0}^{\infty} \gamma^i P(s_i = s)$ is the (unnormalized) discounted visitation frequencies over policy $\boldsymbol{\pi}$. Since the objective in Eq. (5) is difficult to optimize due to the dependency on $\rho_{\boldsymbol{\pi}}(s)$ and $\boldsymbol{\pi}$, we introduce an objective $\hat{\eta}(\boldsymbol{\pi})$ to approximate $\eta(\boldsymbol{\pi})$, instructed by Schulman et al. (2015), as

$$\hat{\eta}(\boldsymbol{\pi}) = \mathbb{E}_{s \sim \rho_{\boldsymbol{\mu}}(s), \boldsymbol{a} \sim \boldsymbol{\pi}(\boldsymbol{a}|s)} \left[ A_{\boldsymbol{\mu}}(s, \boldsymbol{a}) \right]. \tag{6}$$

By replacing the original objective with the surrogate objective, we can formulate the following constrained policy search problem as

$$\arg \max_{\boldsymbol{\pi}} \int_s \rho_{\boldsymbol{\mu}}(s) \int_{\boldsymbol{a}} \boldsymbol{\pi}(\boldsymbol{a}|s) A_{\boldsymbol{\mu}}(s, \boldsymbol{a}) \, d\boldsymbol{a} \, ds, \quad \text{s.t.} \int_s \rho_{\boldsymbol{\mu}}(s) \mathrm{D}_{KL} \left( \boldsymbol{\pi}(\cdot|s) \| \boldsymbol{\mu}(\cdot|s) \right) ds \leq \epsilon.$$

The constraint asserts that when the new policy $\boldsymbol{\pi}$ is close to the basic policy $\boldsymbol{\mu}$, the surrogate objective $\hat{\eta}(\boldsymbol{\pi})$ becomes a precise approximation to $\eta(\boldsymbol{\pi})$[1]. To get the solution to this constrained optimization, we form the Lagrangian of the primal problem presented above,

$$\mathcal{L}(\boldsymbol{\pi}, \beta) = \int_s \rho_{\boldsymbol{\mu}}(s) \int_{\boldsymbol{a}} \boldsymbol{\pi}(\boldsymbol{a}|s) A_{\boldsymbol{\mu}}(s, \boldsymbol{a}) \, d\boldsymbol{a} \, ds + \beta \left( \epsilon - \int_s \rho_{\boldsymbol{\mu}}(s) \mathrm{D}_{\mathrm{KL}} \left( \boldsymbol{\pi}(\cdot|s) \| \boldsymbol{\mu}(\cdot|s) \right) ds \right) \tag{7}$$

---

[1]We refer to Schulman et al. (2015) for a detailed derivation.

where $\beta > 0$ is a Lagrange multiplier.

**Optimal Joint Policy.** According to KKT conditions (Kuhn & Tucker, 1950), the optimal policy $\boldsymbol{\pi}^*$ for the constrained optimization problem in Eq. (7) is expressed by

$$\boldsymbol{\pi}^*(\boldsymbol{a}|s) = \frac{1}{Z(s)}\boldsymbol{\mu}(\boldsymbol{a}|s)\exp\left(\frac{1}{\beta}A_{\boldsymbol{\mu}}(s,\boldsymbol{a})\right), \tag{8}$$

where $Z(s)$ is the partition function.

**Optimal Individual Policy.** Following advantage decomposition in Lemma 1, we can decompose optimal joint policy $\boldsymbol{\pi}^*(\boldsymbol{a}|s)$ to optimal individual policies by assuming the agents choose actions sequentially in the order of $1, 2, ..., n$, as

$$\pi^*(a^i|s, \boldsymbol{a}^{1:i-1}) = \frac{\mu^i(a^i|s, \boldsymbol{a}^{1:i-1})}{Z^i(s)}\exp\left(\frac{1}{\beta}A_{\boldsymbol{\mu}}^i(s,\boldsymbol{a}^{1:i-1},a^i)\right) \tag{9}$$

where $Z^i(s)$ is the partition function. We refer to §A.2 for a detailed derivation of Eqs. (8) and (9).

By maximizing the expected policy improvement $\eta(\boldsymbol{\pi}) = J(\boldsymbol{\pi}) - J(\boldsymbol{\mu})$, we obtain stronger joint and individual policies (i.e., $\boldsymbol{\pi}^*(\boldsymbol{a}|s)$ and $\boldsymbol{\pi}^*(a^i|s, \boldsymbol{a}^{1:i-1})$) over the basic policy $\boldsymbol{\mu} = \boldsymbol{\pi}_{\text{llm}}$. The key insight behind the policy improvement is to re-weight the LLM policy with exponential weights defined in terms of advantages. The advantage function is estimated by local value function $Q_{\boldsymbol{\mu}}^{i_{1:u}}(s, \boldsymbol{a}^{i_{1:u}})$, where we calculate it via Monte-Carlo estimation from a collected dataset $\mathcal{D}$, as we discussed in §3.1.

### 3.3 PROMPTING BY REINFORCED ADVANTAGE FEEDBACK

Upon the basic policy $\boldsymbol{\mu} = \boldsymbol{\pi}_{\text{llm}}$, the advantage-weighted solution in Eq. (9) offers a crucial intuition that (i) by increasing the probability of $\mu^i(a_{\text{pos}}^i|s, \boldsymbol{a}^{1:i-1})$ for those actions $a_{\text{pos}}^i$ with positive advantages, i.e., $A_{\boldsymbol{\mu}}^i(s, \boldsymbol{a}^{1:i-1}, a_{\text{pos}}^i) > 0$, and (ii) decreasing the probability of $\mu^i(a_{\text{neg}}^i|s, \boldsymbol{a}^{1:i-1})$ for those actions $a_{\text{neg}}^i$ with negative advantages, i.e., $A_{\boldsymbol{\mu}}^i(s, \boldsymbol{a}^{1:i-1}, a_{\text{neg}}^i) < 0$, we can ensure an expected performance improvement over $J(\boldsymbol{\mu})$. Therefore, Eq. (9) can be equivalently viewed as behavior cloning (BC) on the *exponential weighting* dataset $\mathcal{D}$ where the better actions are given by higher weights $e^{A_{\boldsymbol{\mu}}^i(s, \boldsymbol{a}^{1:i-1}, a^i)/\beta}$. When $\beta$ is sufficiently small, it becomes BC on a dataset processed by *binary filtering* $\mathbb{1}[A_{\boldsymbol{\mu}}^i(s, \boldsymbol{a}^{1:i-1}, a^i) > 0]$ where $\mathbb{1}$ is the indicator function. This provides an ideal alternative for improving $\boldsymbol{\mu}$ without access to the exact probability of the sampled action $a^i \sim \mu^i(\cdot|s, \boldsymbol{a}^{1:i-1})$, there being convenient for grounding close-source LLMs. We provide theoretical proof for the monotonic improvement with the *binary filtering* in §A.3.

Inspired by the *binary filtering*, we develop a novel feedback mechanism, wherein the main idea is to convert the filter $\mathbb{1}[A_{\boldsymbol{\mu}}^i(s, \boldsymbol{a}^{1:i-1}, a^i) > \epsilon \geq 0]$ into the feedback of LLM-proposed plans with their corresponding scores $A_{\boldsymbol{\mu}}^i(s, \boldsymbol{a}^{1:i-1}, a^i)$ for refining the plans. Based on different types of advantages, we design two algorithms for plan refinement: *ReAd-S* and *ReAd-J*. The process of prompting and refinement is depicted in Figure 2. Algorithmic details of *ReAd-S* and *ReAd-J* are given in §C.

**Prompting and Refinement for *ReAd-S*.** For each time step, we initialize an empty action-set $\boldsymbol{a}_t = \{\}$ and follow the order of $[1, \ldots, n]$ for agents in planning. For planning action $a_t^i$ of agent $i$ at state $s_t$, the process of *ReAd-S* contains two parts. (i) **Prompting as Optimizing.** An LLM planner is given the history of advantages of previous state-action pairs, i.e., $\mathcal{H} = \{(s, (\boldsymbol{a}^{1:i-1}, a^i), A_{\boldsymbol{\mu}}^i(s, \boldsymbol{a}^{1:i-1}, a^i))\}$, and is prompted to *choose an action with the highest advantage* for agent $i$, which recovers the principle of advantage-weighted regression. Leveraging the in-context learning ability, we hope the LLM planner can induce the advantage values of available actions implicitly and choose the action $a_t^i$ with the highest advantage. This process is inspired by recent work for LLM as optimizer (Yang et al., 2023), where the agent is prompted to give a plan that optimizes a score function. (ii) **Feedback for Refinement.** Nevertheless, the implicit advantage maximizing can be hard since the number of available actions can be large. Thus, we introduce a refinement process to allow the LLM to refine the policy if an unsatisfactory action is generated. We use the pre-trained critic network $Q_{\theta}^{i_{1:u}}(s, \boldsymbol{a}^{i_{1:u}})$ with parameter $\theta$ to estimate the advantage score of a generated action, as

$$\mathbb{S}_{\text{ReAd}-\text{S}}(a_t^i) = A_{\theta}^i(s_t, \boldsymbol{a}_t^{1:i-1}, a_t^i) = Q_{\theta}^{1:i}(s_t, \boldsymbol{a}_t^{1:i-1}, a_t^i) - Q_{\theta}^{1:i-1}(s_t, \boldsymbol{a}_t^{1:i-1}).$$

Given a threshold $\epsilon \geq 0$, if the score function is less than the threshold (i.e., $\mathbb{S}_{\text{ReAd}-\text{S}}(a_t^i) < \epsilon$), we add this failed action to the history $\mathcal{H}$ and prompt the agent to re-plan. Such a refinement guarantees

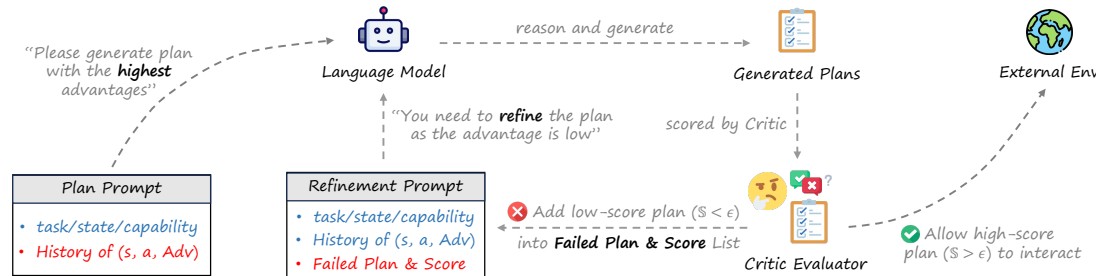

Figure 2: An overview of prompting and refinement. For each timestep $t$, the LLM planner is given the history, which contains states, actions, and advantages, and is prompted to generate a plan with the highest advantage. The pre-trained critic is used to evaluate the score of the generated action $\mathbb{S}_{\mathrm{ReAd}}(a_t^i)$. If $\mathbb{S}_{\mathrm{ReAd}}(a_t^i) < \epsilon$, the failed plan is used as a prompt, and the LLM planer is asked to refine the policy until the $\mathbb{S}_{\mathrm{ReAd}}(a_t^i) > \epsilon$. The (refined) action is used to interact with the environment, and the LLM planner is processed in the next step.

embodied agents always take the actions with $A_\theta^i(s_t, \boldsymbol{a}_t^{1:i-1}, a_t^i) > \epsilon$, further ensuring monotonic improvements over $\boldsymbol{\pi}_{\mathrm{llm}}$. It significantly decreases the interaction rounds of agents since the action $a_t^i$ has been evaluated and refined via advantage feedback before execution. In contrast, previous methods like RoCo need to interact with the environment to get physical feedback regardless of the quality of the generated actions. The refined action is added into the action-set $\boldsymbol{a}_t \leftarrow \boldsymbol{a}_t \cup \{a_t^i\}$ and we then perform sequential decision for agent $i + 1$.

**Prompting and Refinement for *ReAd-J*.** The planning process of the LLM planner for *ReAd-J* is similar to that of *ReAd-S*. The main difference is the LLM planner for *ReAd-J* is required to give a joint action $\boldsymbol{a}_t$ for all agents at once. Meanwhile, we use the joint advantage function for history prompting with $\mathcal{H} = \{(s, \boldsymbol{a}_t, A_{\boldsymbol{\mu}}(s_t, \boldsymbol{a}_t))\}$ rather than considering the local advantages. The score function is

$$\mathbb{S}_{\mathrm{ReAd-J}}(\boldsymbol{a}_t) = A_\theta(s_t, \boldsymbol{a}_t) = Q_\theta(s_t, \boldsymbol{a}_t) - \frac{1}{\gamma} Q_\theta(s_t, \boldsymbol{w})$$

based on Eq. (8). The joint plan $\boldsymbol{a}_t$ is refined if it is less than a threshold (i.e., $\mathbb{S}_{\mathrm{ReAd-J}}(\boldsymbol{a}_t) < \epsilon$).

## 4 RELATED WORKS

**Task Planning with LLMs.** LLMs (Chowdhery et al., 2023; OpenAI, 2023; Touvron et al., 2023a;b) trained on a large-scale corpus exhibits notable reasoning abilities via in-context learning (Dong et al., 2022; Abernethy et al., 2023; Akyürek et al., 2023). However, LLMs can also give infeasible plans for embodied agents due to the lack of real-world knowledge. A line of research modifies the open-loop planning framework to a closed-loop one via self-evaluation and reflection. For example, ReAct (Yao et al., 2023b), Reflexion (Shinn et al., 2023), and BeamSearch (Xie et al., 2023b) incorporate the feedback of an LLM evaluator in the prompts after the previous plan is completed. Other works integrate domain knowledge of embodied agents in feedback. For example, RoCo (Mandi et al., 2023) and Inner Monologue (Huang et al., 2022b) design physical verification such as collision checking, object recognition, and scene description for feedback. DoReMi (Guo et al., 2023) leverages LLM to generate physical constraints, and ViLA (Hu et al., 2023b) adopts Vision-Language Model (VLM) as a constraint detector for verification. Another line of research develops advanced reasoning frameworks, including chain-of-thought (Wei et al., 2022; Mu et al., 2023) and tree-of-thought (Yao et al., 2023a). Works like (Zhao et al., 2023; Hao et al., 2023) consider LLMs as a world model (Lin et al., 2023c) and adopt tree search in planning (Hu et al., 2023a). Other works adopt the planning domain definition language (PDDL) for searching in long-horizon problems (Silver et al., 2023; Liu et al., 2023a; Zhou et al., 2023). Our work lies in closed-loop frameworks but has a novel advantage function in feedback, which is different from self-reflection or physical feedback and does not rely on advanced searching algorithms.

**Grounding LLM with RL.** RL with Human Feedback (RLHF) has been used to align LLM with human preference through parameter tuning (Dai et al., 2023; Fernandes et al., 2023; Song et al., 2023b). In contrast, our work focuses on grounding closed-source LLM with RL via few-shot prompting and closed-loop feedback (Zeng et al., 2023; Wu et al., 2023; Huang et al., 2022a; Lin

et al., 2023b). Previous works tried to integrate RL into LLM planning under the framework tree search (Browne et al., 2012). For example, FAFA (Liu et al., 2023b) and TS-LLM (Feng et al., 2023) learn an environment model and value function to plan the subroutine in MCTS. REX (Murthy et al., 2023) proposes to balance exploration and exploitation in LLM-based MCTS. Other works like SayCan (Ahn et al., 2022) and Text2Motion (Lin et al., 2023d) adopt a model-free manner by learning value functions to connect LLM knowledge to physical environments. SwiftSage (Lin et al., 2023a) performs imitation learning for rapid thinking and LLM for methodical training. Remember (Zhang et al., 2023b) learns value functions for LLM to predict $Q$-value via exemplars in prompts and select actions based on $Q$-values. Unlike the Remember framework, which retrieves similar states from a buffer, we evaluate the advantage function of planned actions via a neural network and follow advantage-weighted regression in prompting. We employ the advantage function in a multi-agent setting, while previous methods focus on single-agent planning. Previous LLM-based multi-agent works mostly manually designed communication, reflection, and reasoning modules (Zhang et al., 2023a;c; Kannan et al., 2023; Chen et al., 2023). CAMEL (Li et al., 2023a) facilitated cooperation among communicative agents through role-playing and inception prompting, which also includes a critic with different purposes and does not have theoretical guarantees. MetaGPT (Hong et al., 2023) similarly incorporated Standardized Operating Procedures (SOPs) into LLM-based multi-agent collaborations where the roles of each agent was predefined by humans. Compared to previous LLM-based multi-agent works, we propose a more principled way by using the sequential advantage function from multi-agent RL for cooperation.

## 5 EXPERIMENTS

We first introduce two multi-agent collaboration environment in §5.1. Then we design a series of experiments to compare our approach with baselines in §5.2. Finally, we conduct ablation studies and analyze the impact of modules in §5.3.

### 5.1 EXPERIMENTAL SETUP

**DV-RoCoBench.** We present *Difficult Variants of RoCoBench (DV-RoCoBench)* for embodied multi-robot collaboration, which is derived from RoCoBench (Mandi et al., 2023). RoCoBench consists of 6 multi-robot collaboration tasks in a tabletop manipulation environment, typically involving interactive objects that are semantically straightforward to comprehend and reason about for LLMs. The tasks encompass a range of collaboration scenarios that necessitate robots' communication and coordination behaviors. Robots receive their observation and select one action from the high-level action set, which includes diverse functionalities such as WAIT, moving, sweeping, grasping, and dropping, across multiple tasks. The execution of high-level actions is subsequently translated into low-level actions for manipulation. In contrast to RoCoBench, which focuses primarily on tasks with a fixed difficulty level, we select three tasks to enrich the complexity of the benchmark and create the new *DV-RoCoBench*, where each task is tailored to have 4-5 difficulty levels for experiments. Due to technically unresolved issue in the original RoCoBench, we have already selected all executable tasks to form our newly developed *DV-RoCoBench*.

In the following, we give a brief description of tasks and settings. See §D for details.

- **Sweep Floor.** Two robot arms need to work together to sweep all the cubes on the table into the bin. The aim is to sweep away the cubes with given colors. We establish 5 difficulty levels based on the number of overall cubes and the target cubes. An LLM planner is more likely to produce fact hallucinations in more difficult settings.
- **Make Sandwich.** Two robot arms need to stack the ingredients to make a sandwich according to the recipe. Each arm is limited in operating range and cooperation between agents is required. We establish 4 difficulty levels depending on the length of the recipe.
- **Sort Cubes.** Three robot arms within their operating ranges are required to coordinate and place cubes on the table to their target positions. We establish 5 different difficulty levels based on the distance between the cubes and their target locations.

**Overcooked-AI.** *Overcooked-AI* (Carroll et al., 2019) is a fully cooperative multi-agent benchmark environment based on the wildly popular video game Overcooked. In this environment, agents need to deliver soups as fast as possible. Each soup requires placing up to 3 ingredients in a pot,

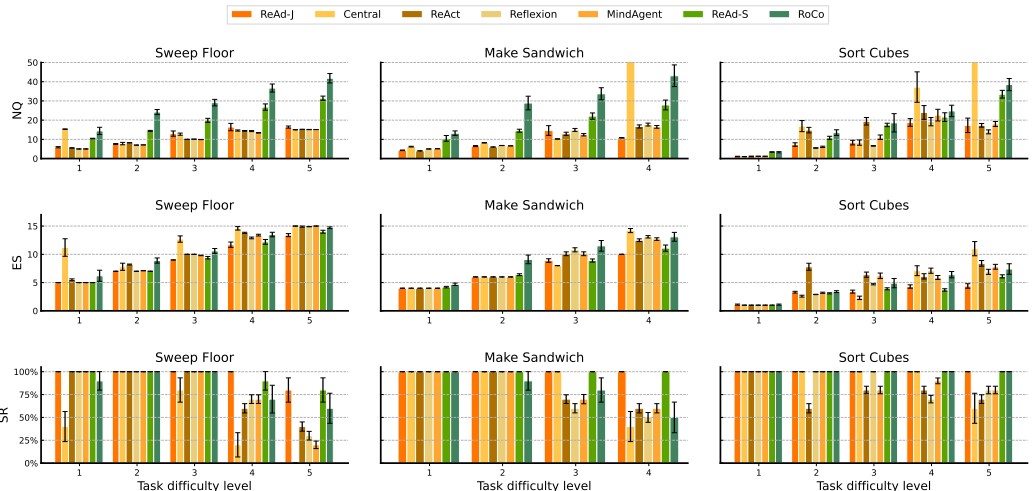

Figure 3: We report mean SR (↑), ES (↓), and NQ (↓) in 3 tasks with various difficulty levels averaged over 10 random seeds. The detailed score is given in Table 4 of §E.2.

waiting for the soup to cook, and having an agent pick up the soup and deliver it. The environment consists of 5 different kitchen scenarios, covering from low-level motion coordination challenges to high-level strategy coordination challenges. In our experiment, we chose two representative scenarios: **Cramped Room** and **Forced Coordination**, and set the number of ingredients to make soups as 2 and the timesteps to cook as 2. To enable the computation of the success rate, we modify the task to cook and deliver a soup within a specified number of timesteps. Details of the environment are given in §D.4. For quantitative comparisons, we impose the maximum number of environment steps per episode to 15 in *DV-RoCoBench*, 20 in Cramped Room, and 25 in Forced Coordination. Specially, for our adapted **Cramped Room** and **Forced Coordination**, we deliberately set the maximum environment steps almost equal to the least number of environment steps for accomplishing the task, thereby presenting a challenge for highly effective coordination. And the maximum rounds of re-planning per step is set to 15 for all tasks except for Sort Cubes where it is set to 10.

**Baseline Methods.** We use GPT-4-Turbo (OpenAI, 2023) as the basic LLM policy for all experiments. Since our *ReAd* lies in the setting of LLM grounding on embodied tasks, we mainly choose LLM-based methods as baselines. On both benchmarks, we compare *ReAd-J* with three strong close-loop baselines – ReAct (Yao et al., 2023b), Reflexion (Shinn et al., 2023) and MindAgent (Gong et al., 2023), and a planner named Central Plan which instructs the LLM to generate actions for all robots based on the history of all agents. These five methods output agents' plans in a parallel manner. In *DV-RoCoBench*, we particularly add one more baseline RoCo (Mandi et al., 2023) which achieves the state-of-the-art performance in RoCoBench (Mandi et al., 2023), for comparisons with *ReAd-S*. Both of them generate joint plans in a sequential manner. Due to the expensive cost of sequential planning with more environment steps in *Overcooked-AI*, we only evaluate the performance of methods that generate joint plans in a parallel manner. We provide a detailed comparison in Table 3 of §E.1.

**Evaluation Metrics.** We evaluate the performance of algorithms on three metrics that closely resemble that in RoCoBench: (i) **SR**: the success rate of completing tasks within the limited interaction rounds; (ii) **ES**: the number of interaction steps to the environment taken by the robots to complete the task; (iii) **NQ**: the number of queries to LLMs in completing the task, which measures the efficiency in enquiring LLMs to obtain a feasible plan. An algorithm is better if it has *higher SR, fewer ES, and fewer NQ*. Among these metrics, SR and ES directly reflect the effectiveness of a planner in completing tasks, while NQ can be somewhat trivial since a planner can have much fewer queries to LLM but has a low SR. In contrast, methods that require policy refinement often require more queries to lead to a high SR.

## 5.2 RESULTS

***ReAd-S* and *ReAd-J* outperform their corresponding strong baselines on all metrics and achieve more efficient LLM grounding.** As shown in Figure 3, with the increase of difficulty levels in *DV-RoCoBench*, the performance contrast in SR becomes pronounced gradually. In more difficult

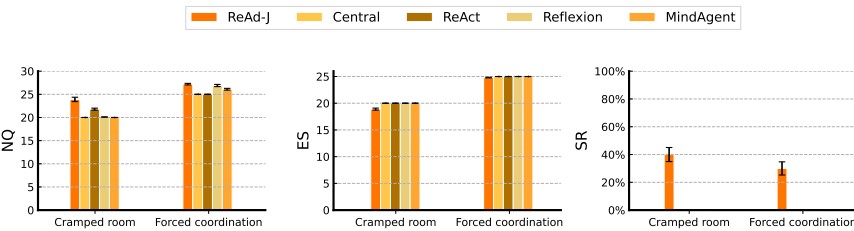

Figure 4: We report mean SR (↑), ES (↓), and NQ (↓) in two scenarios of *Overcooked-AI* averaged over 10 random seeds. The detailed score is given in Table 5 of §E.2.

settings (e.g., level 4 or 5 in tasks), our approach obtains higher success rates while baseline methods fail to make progress. Meanwhile, *ReAd-S* and *ReAd-J* present lower ES and comparable or even lower NQ on most tasks in *DV-RoCoBench* when compared to their corresponding baselines. A lower ES suggests that prompting LLMs to generate actions maximizing the advantages can improve the optimality of the proposed plans because a higher advantage implies the generated action contributes more to accomplishing the task. Furthermore, as shown in Figure 4, our methods achieve a significantly higher SR compared with the methods relying on *physical verification* as feedback in *Overcooked-AI*. Due to the heavy coordination challenges inherent to *Overcooked-AI*, LLM-based agents cannot advance toward task completion unless the LLM planner generates highly collaborative plans. By replacing the *physical verification* feedback with *advantage function*, we implicitly transfer the understanding and reasoning of the LLMs from semantic comprehension towards the current state of the environment to digesting the numerical relationship. As the scenario becomes more challenging for multi-agent collaboration, it is inevitable to involve more redundant information and disturbing components in the environment, which poses a challenge for the LLM planner to capture and reason about the essential part inside the state and physical feedback. In contrast, benefiting from *ReAd* feedback, the LLM planner only needs to concentrate on how to maximize the advantage score no matter how challenging the scenario is. Hence, our approach exhibits superior planning capabilities and better LLM grounding results for embodied tasks. Additionally, we evaluate the performance of the open-source model Llama-3.1-70B-Instruct (Dubey et al., 2024) equipped with our algorithm on the *Y2_G3* task. The result is provided in §E.3.

**With sudden disturbances towards the environments, the LLM-planner can re-adjust plans rapidly to accomplish the task via *ReAd* feedback.** Since the critic takes both the current state and the proposed actions as input, it endows the LLM planner with not only the foresight to discern whether the action contributes to realizing the goal but also the ability to reschedule the planning quickly when encountering sudden disturbances to the advancement of the task. To evaluate the robustness of the LLM planner, we compare *ReAd-S* and RoCo in extra extended scenarios with unexpected disruptions. We select *recipe3* (3rd difficulty level in Make Sandwich) that takes a minimum environment step of 8 to accomplish the task. When a disruption occurs at timestep $n$ ($0 \leq n < 8, n \in \mathbb{N}$), we reset the task and reinitialize the state without giving any hints about this resetting in the prompt and clearing previous history information contained in the prompt. Specifically, the "adversarial" case affects the LLM-based agent from two aspects: (i) the description of current state $s_{\text{reset}}$ which is given to the LLM planner before planning; (ii) the unexpected transition of environment after executing an action. It raises an intractable challenge as the remaining historical information becomes misaligned with the actual situation. The lack of a complete description of the sudden disruption significantly increases the likelihood of the LLM planner proposing erroneous actions.

Table 1: Evaluation results over 10 runs of *ReAd-S* and RoCo and its modified versions on disturbances at timestep $n$. We present the disturbance as resetting the environment. $n = 0$: no resetting.

| | Method | NQ | ES | SR |
|---|---|---|---|---|
| recipe3 ($n=0$) | ReAd-S | 22.1±1.65 | 8.9±0.28 | 1.0±0.00 |
| | RoCo-L | 44.7±4.90 | 12.0±0.54 | 0.9±0.10 |
| | RoCo-P | 33.7±3.16 | 11.5±0.95 | 0.8±0.13 |
| | RoCo | 33.7±3.16 | 11.5±0.95 | 0.8±0.13 |
| recipe3 ($n=1$) | ReAd-S | 39.7±5.30 | 10.4±0.34 | 1.0±0.00 |
| | RoCo-L | 55.3±2.63 | 14.1±0.28 | 0.8±0.13 |
| | RoCo-P | 33.6±2.03 | 12.5±0.73 | 0.9±0.10 |
| | RoCo | 46.3±3.60 | 13.9±0.43 | 0.7±0.15 |
| recipe3 ($n=2$) | ReAd-S | 44.9±4.34 | 12.5±0.34 | 1.0±0.00 |
| | RoCo-L | 53.4±2.28 | 14.8±0.20 | 0.3±0.15 |
| | RoCo-P | 35.2±0.98 | 14.3±0.26 | 0.8±0.13 |
| | RoCo | 61.2±11.95 | 14.2±0.44 | 0.5±0.16 |
| recipe3 ($n=3$) | ReAd-S | 49.1±4.53 | 13.4±0.54 | 1.0±0.00 |
| | RoCo-L | 75.9±6.91 | 15.0±0.00 | 0.0±0.00 |
| | RoCo-P | 40.0±2.94 | 14.3±0.26 | 0.5±0.17 |
| | RoCo | 74.8±10.79 | 15.0±0.00 | 0.0±0.00 |

To eliminate the influence induced by the different history information utilized between *ReAd-S*

and RoCo, we provide two more variants of RoCo as baselines. One uses only the history of the previous round, which we name RoCo-L, while the other is informed with descriptions of the sudden disturbance, which we name RoCo-P. The evaluation results are shown in Table 1. A larger step $n$ signifies a more severe influence of disturbance. As $n$ increases from 0 to 3, *ReAd-S* consistently outperforms RoCo and its variants on SR and ES. Although RoCo retains a high SR under $n = 1, 2$, it fails to recalibrate the misalignment between the remaining history information and the actual status of the environment, leading to a significant drop in SR when $n = 3$. Regardless of what kind of history information RoCo relies on, consistent superior performance demonstrates that *ReAd* feedback alleviates the potentially severe hallucination issue and brings reliable robustness.

## 5.3 Ablation Studies

**Plan refinement has a remarkable impact on grounding LLM.** The advantage score plays two roles in ReAd: (i) *prompting as optimizing* for generating actions with the highest score, and (ii) *feedback as refinement* for re-plan if the score is less than a threshold. The policy refinement makes our method a *multi-step* process since the action can be refined for multi-rounds. To investigate the role of plan refinement, we adopt a *single-step* version by removing the second role, which forms an open-loop plan genera-

Table 2: The performance of the multi-step and single-step version of *ReAd-S* and *ReAd-J* on the *Y3_G3* task.

|  | NQ | ES | SR |
| --- | --- | --- | --- |
| ReAd-J(Multi-Step) | 16.4±0.54 | 13.4±0.27 | 0.8±0.13 |
| ReAd-J(Single-Step) | 19.1±1.25 | 14.1±0.28 | 0.6±0.16 |
| ReAd-S(Multi-Step) | 31.4±1.11 | 14.0±0.26 | 0.8±0.13 |
| ReAd-S(Single-Step) | 35.1±1.16 | 14.5±0.17 | 0.6±0.16 |

tion without refinement. In Table 2, we denote the original version as *Multi-Step* and the open-loop version as *Single-Step*. We pick the most difficult variant *Y3_G3* in Sweep Floor and observe a marginal decline in both efficiency and success rates in *Single-Step*. It suggests that plan refinement that ensures monotonic policy improvement is crucial for performance. Interestingly, *ReAd-J(Single-Step)* can also achieve a considerable success rate of 60%, which is dramatically comparable or superior to the baselines with *physical verification* as feedback.

## 6 Discussion and Conclusion

We have presented *ReAd* as a novel LLM feedback for closed-loop planning in multi-agent collaboration. We provide theoretical motivation based on multi-agent advantage-weighted regression. The LLM is prompted to generate plans with high advantages and perform policy refinement. The experiments on *DV-RoCoBench* and *Overcooked-AI* show that our method outperforms physical feedback with improved efficiency. Moreover, the advantage feedback can handle sudden disturbances and is crucial for refinement. Due to the limitation of currently available benchmark for embodied multi-agent collaboration evaluation, most of our experiments are conducted in 2 or 3-agent scenarios. In a case with an increasing number of agents, theoretically speaking, *ReAd-J* would be hindered by the exponential growth of the joint state-action space while *ReAd-S* could maintain consistent performance by scoring in the individual state-action space, enjoying the benefit of sequential decision-making manner. However, it also necessitates more computational costs and time for dataset collection in such a scenario. Thus, how our proposed *ReAd* feedback mechanism practically scales under scenarios with many agents remains fascinating. To this end, building a well-established embodied many-agent collaboration benchmark is essential, which provides an opportunity to push our algorithm to the limit. We consider investigating the *ReAd* feedback mechanism in the many-agent scenario and tackling the potential limitation in future works. Future works also include extending the advantage feedback to multi-objective and safe planning scenarios. Last but not least, we provide extended discussion on Symbol Grounding Problem (Harnad, 1990) in §F.

## Reproducibility Statement

For the theoretical motivation of multi-agent advantages, we provide the detailed theoretical proof in Appendix A. The experiment setup and implementation details are given in Appendix D. The prompts, interaction process of LLMs, and videos of interaction process are provided in the Appendix E, Appendix H, and the project website `https://read-llm.github.io`. The code will be released publicly after the review process.

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

## A  THEORETICAL PROOF

### A.1  PROOF OF MULTI-AGENT ADVANTAGE DECOMPOSITION

*Proof.* With the definition of the multi-agent local advantage function in Eq. (3), we can have

$$
\begin{aligned}
\sum_{k=1}^{n} A_{\boldsymbol{\pi}}^{i_k}(s, \boldsymbol{a}^{i_{1:k-1}}, a^{i_k}) &= \sum_{k=1}^{n} Q_{\boldsymbol{\pi}}^{i_{1:k}}(s, \boldsymbol{a}^{i_{1:k}}) - Q_{\boldsymbol{\pi}}^{i_{1:k-1}}(s, \boldsymbol{a}^{i_{1:k-1}}) \\
&= Q_{\boldsymbol{\pi}}^{i_{1:n}}(s, \boldsymbol{a}^{i_{1:n}}) - Q_{\boldsymbol{\pi}}^{i_{1:n-1}}(s, \boldsymbol{a}^{i_{1:n-1}}) + Q_{\boldsymbol{\pi}}^{i_{1:n-1}}(s, \boldsymbol{a}^{i_{1:n-1}}) - Q_{\boldsymbol{\pi}}^{i_{1:n-2}}(s, \boldsymbol{a}^{i_{1:n-2}}) \\
&\quad + ... + Q_{\boldsymbol{\pi}}^{i_{1:1}}(s, \boldsymbol{a}^{i_{1:1}}) - Q_{\boldsymbol{\pi}}^{i_{1:0}}(s, \boldsymbol{a}^{i_{1:0}}) \\
&= Q_{\boldsymbol{\pi}}^{i_{1:n}}(s, \boldsymbol{a}^{i_{1:n}}) - Q_{\boldsymbol{\pi}}^{i_{1:0}}(s, \boldsymbol{a}^{i_{1:0}}) \\
&= Q_{\boldsymbol{\pi}}(s, \boldsymbol{a}) - V_{\boldsymbol{\pi}}(s) \\
&= A_{\boldsymbol{\pi}}(s, \boldsymbol{a}).
\end{aligned}
$$

$\square$

### A.2  DERIVATION OF OPTIMAL JOINT POLICY AND OPTIMAL INDIVIDUAL POLICY

In this section, we begin with the constrained policy search problem. Following the performance difference lemma (Kakade & Langford, 2002), the expected improvement $\eta(\boldsymbol{\pi}) = J(\boldsymbol{\pi}) - J(\boldsymbol{\mu})$ can be expressed by

$$
\begin{aligned}
\mathbb{E}_{s_0, \boldsymbol{a}_0, ... \sim \boldsymbol{\pi}} \left[ \sum_{t=0}^{\infty} \gamma^t A_{\boldsymbol{\mu}}(s_t, \boldsymbol{a}_t) \right] &= \mathbb{E}_{s_0, \boldsymbol{a}_0, ... \sim \boldsymbol{\pi}} \left[ \sum_{t=0}^{\infty} \gamma^t \left( r(s_t, \boldsymbol{a}_t) + \gamma V_{\boldsymbol{\mu}}(s_{t+1}) - V_{\boldsymbol{\mu}}(s_t) \right) \right] \\
&= \mathbb{E}_{s_0, \boldsymbol{a}_0, ... \sim \boldsymbol{\pi}} \left[ -V_{\boldsymbol{\mu}}(s_0) + \sum_{t=0}^{\infty} \gamma^t r(s_t, \boldsymbol{a}_t) \right] \\
&= -\mathbb{E}_{s_0 \sim p(s_0)} \left[ V_{\boldsymbol{\mu}}(s_0) \right] + \mathbb{E}_{s_0, \boldsymbol{a}_0, ... \sim \boldsymbol{\pi}} \left[ \sum_{t=0}^{\infty} \gamma^t r(s_t, \boldsymbol{a}_t) \right] \\
&= -J(\boldsymbol{\mu}) + J(\boldsymbol{\pi}).
\end{aligned}
\tag{10}
$$

We can rewrite Eq. (10) with an expectation over states using discounted visitation frequencies $\rho_{\boldsymbol{\pi}}(s)$,

$$
\begin{aligned}
\eta(\boldsymbol{\pi}) &= \mathbb{E}_{s_0, \boldsymbol{a}_0, ... \sim \boldsymbol{\pi}} \left[ \sum_{t=0}^{\infty} \gamma^t A_{\boldsymbol{\mu}}(s_t, \boldsymbol{a}_t) \right] \\
&= \sum_{t=0}^{\infty} \int_s p(s_t = s | \boldsymbol{\pi}) \int_{\boldsymbol{a}} \boldsymbol{\pi}(\boldsymbol{a}|s) \gamma^t A_{\boldsymbol{\mu}}(s, \boldsymbol{a}) \, d\boldsymbol{a} \, ds \\
&= \int_s \sum_{t=0}^{\infty} \gamma^t p(s_t = s | \boldsymbol{\pi}) \int_{\boldsymbol{a}} \boldsymbol{\pi}(\boldsymbol{a}|s) A_{\boldsymbol{\mu}}(s, \boldsymbol{a}) \, d\boldsymbol{a} \, ds \\
&= \int_s \rho_{\boldsymbol{\pi}}(s) \int_{\boldsymbol{a}} \boldsymbol{\pi}(\boldsymbol{a}|s) A_{\boldsymbol{\mu}}(s, \boldsymbol{a}) \, d\boldsymbol{a} \, ds,
\end{aligned}
\tag{11}
$$

where $\rho_{\boldsymbol{\pi}}(s) = \sum_{t=0}^{\infty} \gamma^t p(s_t = s | \boldsymbol{\pi})$ represents the (unnormalized) discounted visitation frequencies over policy $\boldsymbol{\pi}$ and $p(s_t = s | \boldsymbol{\pi})$ is the likelihood of the agent at state $s$ after following $\boldsymbol{\pi}$ for $t$ timesteps. Our goal is to find the optimal policy $\boldsymbol{\pi}^*$ that maximizes the expected improvement $\eta(\boldsymbol{\pi})$.

However, it's intractable to sample over the target policy $\boldsymbol{\pi}$, further causing that the objective in Eq. (11) can be difficult to optimize. Following (Schulman et al., 2015), we can introduce an approximation $\hat{\eta}(\boldsymbol{\pi})$ of $\eta(\boldsymbol{\pi})$ using the discounted visitation frequencies over the old policy $\boldsymbol{\mu}$,

$$
\hat{\eta}(\boldsymbol{\pi}) = \int_s \rho_{\boldsymbol{\mu}}(s) \int_{\boldsymbol{a}} \boldsymbol{\pi}(\boldsymbol{a}|s) A_{\boldsymbol{\mu}}(s, \boldsymbol{a}) \, d\boldsymbol{a} \, ds.
$$

$\hat{\eta}(\boldsymbol{\pi})$ matches $\eta(\boldsymbol{\pi})$ to first order (Kakade & Langford, 2002), and provides a good estimate of $\eta$ if $\boldsymbol{\pi}$ is close enough to $\boldsymbol{\mu}$. In practice, we initialize the target policy $\boldsymbol{\pi}$ with the LLM policy $\boldsymbol{\mu}$ to satisfy

the above condition. Therefore, we can formulate the following constrained policy search problem,

$$\arg\max_{\boldsymbol{\pi}} \quad \int_s \rho_{\boldsymbol{\mu}}(s) \int_{\boldsymbol{a}} \boldsymbol{\pi}(\boldsymbol{a}|s) A_{\boldsymbol{\mu}}(s, \boldsymbol{a}) \, d\boldsymbol{a} \, ds, \tag{12}$$

$$\text{s.t.} \quad \mathrm{D_{KL}}\left(\boldsymbol{\pi}(\cdot|s)\|\boldsymbol{\mu}(\cdot|s)\right) \leq \epsilon, \quad \forall s, \tag{13}$$

$$\int_{\boldsymbol{a}} \boldsymbol{\pi}(\boldsymbol{a}|s) \, d\boldsymbol{a} = 1, \quad \forall s. \tag{14}$$

However, enforcing the pointwise KL constraint in Eq. (13) at all states is intractable. To simplify the constrained optimization problem, we relax the hard KL constraint by converting it into a soft constraint in an expectation form, as

$$\arg\max_{\boldsymbol{\pi}} \quad \int_s \rho_{\boldsymbol{\mu}}(s) \int_{\boldsymbol{a}} \boldsymbol{\pi}(\boldsymbol{a}|s) A_{\boldsymbol{\mu}}(s, \boldsymbol{a}) \, d\boldsymbol{a} \, ds,$$

$$\text{s.t.} \quad \int_s \rho_{\boldsymbol{\mu}}(s) \mathrm{D_{KL}}\left(\boldsymbol{\pi}(\cdot|s)\|\boldsymbol{\mu}(\cdot|s)\right) \, ds \leq \epsilon,$$

$$\int_{\boldsymbol{a}} \boldsymbol{\pi}(\boldsymbol{a}|s) \, d\boldsymbol{a} = 1, \quad \forall s.$$

Next, we form the Lagrangian, as

$$\mathcal{L}(\boldsymbol{\pi}, \beta, \nu) = \int_s \rho_{\boldsymbol{\mu}}(s) \int_{\boldsymbol{a}} \boldsymbol{\pi}(\boldsymbol{a}|s) A_{\boldsymbol{\mu}}(s, \boldsymbol{a}) \, d\boldsymbol{a} \, ds + \beta \left( \epsilon - \int_s \rho_{\boldsymbol{\mu}}(s) \mathrm{D_{KL}}\left(\boldsymbol{\pi}(\cdot|s)\|\boldsymbol{\mu}(\cdot|s)\right) \, ds \right)$$

$$+ \int_s \nu_s \left( 1 - \int_a \boldsymbol{\pi}(\boldsymbol{a}|s) \, d\boldsymbol{a} \right) \, ds,$$

where $\nu = \{\nu_s | \forall s \in \mathcal{S}\}$ and $\beta > 0$ correspond to the Lagrange multipliers.

**Derivation of Optimal Joint Policy.** Differentiating $\mathcal{L}(\boldsymbol{\pi}, \beta, \nu)$ with respect to $\boldsymbol{\pi}(\boldsymbol{a}|s)$ gives the following,

$$\frac{\partial \mathcal{L}}{\partial \boldsymbol{\pi}(\boldsymbol{a}|s)} = \rho_{\boldsymbol{\mu}}(s) A_{\boldsymbol{\mu}}(s, \boldsymbol{a}) - \beta \rho_{\boldsymbol{\mu}}(s) \log \boldsymbol{\pi}(\boldsymbol{a}|s) + \beta \rho_{\boldsymbol{\mu}}(s) \log \boldsymbol{\mu}(\boldsymbol{a}|s) - \beta \rho_{\boldsymbol{\mu}}(s) - \nu_s. \tag{15}$$

According to KKT conditions (Kuhn & Tucker, 1950), if $(\boldsymbol{\pi}^*, \beta^*, \nu^*)$ is a saddle point of $\mathcal{L}$, $\boldsymbol{\pi}^*$ is the optimal solution of the primal problem. Thus, let Eq. (15) be equal to zero, then we have

$$\log \boldsymbol{\pi}^*(\boldsymbol{a}|s) = \frac{1}{\beta^*} A_{\boldsymbol{\mu}}(s, \boldsymbol{a}) + \log \boldsymbol{\mu}(\boldsymbol{a}|s) - 1 - \frac{1}{\rho_{\boldsymbol{\mu}}(s)} \frac{\nu_s^*}{\beta^*}, \tag{16}$$

$$\boldsymbol{\pi}^*(\boldsymbol{a}|s) = \boldsymbol{\mu}(\boldsymbol{a}|s) \exp\left( \frac{1}{\beta^*} A_{\boldsymbol{\mu}}(s, \boldsymbol{a}) \right) \exp\left( -\frac{1}{\rho_{\boldsymbol{\mu}}(s)} \frac{\nu_s^*}{\beta^*} - 1 \right). \tag{17}$$

Note that the primal problem holds the constraint $\int_{\boldsymbol{a}} \boldsymbol{\pi}(\boldsymbol{a}|s) \, d\boldsymbol{a} = 1$, the second exponential term is consequently viewed as the partition function $Z(s)$ that normalizes the conditional action distribution,

$$Z(s) = \exp\left( \frac{1}{\rho_{\boldsymbol{\mu}}(s)} \frac{\nu_s^*}{\beta^*} + 1 \right) = \int_{\boldsymbol{a}'} \boldsymbol{\mu}(\boldsymbol{a}'|s) \exp\left( \frac{1}{\beta^*} A_{\boldsymbol{\mu}}(s, \boldsymbol{a}') \right) \, d\boldsymbol{a}'. \tag{18}$$

*Optimal Joint Policy* is then given by,

$$\underbrace{\boldsymbol{\pi}^*(\boldsymbol{a}|s)}_{\text{Left-Hand Side}} = \underbrace{\frac{1}{Z(s)} \boldsymbol{\mu}(\boldsymbol{a}|s) \exp\left( \frac{1}{\beta^*} A_{\boldsymbol{\mu}}(s, \boldsymbol{a}) \right)}_{\text{Right-Hand Side}}. \tag{19}$$

**Derivation of Optimal Individual Policy.** Given the set of agents $\mathcal{N} = \{1, 2, ..., n\}$, we assume the agents choose actions sequentially in the order of $1, 2, ..., n$, i.e., agents $i$ is aware of current state $s$ and the chosen actions of agents $1, 2, ..., i-1$ and select actions based on that. The following equation holds by the support of the definition of conditional probability,

$$\boldsymbol{\pi}(\boldsymbol{a}|s) = \prod_{i=1}^{n} \pi^i(a^i|s, \boldsymbol{a}^{1:i-1}), \tag{20}$$

where $\pi^i$ is the individual policy of agent $i$. Here we consider a general case that the old joint policy and the target joint policy are both in a sequential manner. Following multi-agent advantage decomposition in Lemma 1, the LHS and RHS of Eq. (19) can be expressed respectively (in order to present the *Optimal Individual Policy* we omit the superscript of it which denotes agent id),

$$\text{LHS} = \prod_{i=1}^{n} \pi^*(a^i|s, \boldsymbol{a}^{1:i-1}), \tag{21}$$

$$\text{RHS} = \frac{1}{Z(s)} \prod_{i=1}^{n} \mu^i(a^i|s, \boldsymbol{a}^{1:i-1}) \exp\left(\frac{1}{\beta^*} A_{\boldsymbol{\mu}}^i(s, \boldsymbol{a}^{1:i-1}, a^i)\right)$$

$$= \prod_{i=1}^{n} \frac{1}{Z^i(s)} \mu^i(a^i|s, \boldsymbol{a}^{1:i-1}) \exp\left(\frac{1}{\beta^*} A_{\boldsymbol{\mu}}^i(s, \boldsymbol{a}^{1:i-1}, a^i)\right). \tag{22}$$

Thus, we can get the expression of *Optimal Individual Policy*,

$$\pi^*(a^i|s, \boldsymbol{a}^{1:i-1}) = \frac{1}{Z^i(s)} \mu^i(a^i|s, \boldsymbol{a}^{1:i-1}) \exp\left(\frac{1}{\beta^*} A_{\boldsymbol{\mu}}^i(s, \boldsymbol{a}^{1:i-1}, a^i)\right), \tag{23}$$

where $Z^i(s)$ is the partition function that normalizes the conditional action distribution $\pi^*(a^i|s, \boldsymbol{a}^{1:i-1})$ of agent $i$ and satisfies $Z(s) = \prod_{i=1}^{n} Z^i(s)$. Finally, all that remains for us to do is to derive the validity of $Z(s) = \prod_{i=1}^{n} Z^i(s)$.

Since $Z^i(s)$ is the partition function that normalizes the conditional action distribution $\pi^*(a^i|s, \boldsymbol{a}^{1:i-1})$, we can have,

$$Z^i(s) = \int_{a^i} \mu^i(a^i|s, \boldsymbol{a}^{1:i-1}) \exp\left(\frac{1}{\beta^*} A_{\boldsymbol{\mu}}^i(s, \boldsymbol{a}^{1:i-1}, a^i)\right) da^i. \tag{24}$$

Meanwhile, we can rewrite Eq. (18) after applying multi-agent advantage decomposition in Lemma 1,

$$Z(s) = \int_{\boldsymbol{a}} \boldsymbol{\mu}(\boldsymbol{a}|s) \exp\left(\frac{1}{\beta^*} A_{\boldsymbol{\mu}}(s, \boldsymbol{a})\right) d\boldsymbol{a} \tag{25}$$

$$= \prod_{i=1}^{n} \int_{a^i} \mu^i(a^i|s, \boldsymbol{a}^{1:i-1}) \exp\left(\frac{1}{\beta^*} A_{\boldsymbol{\mu}}^i(s, \boldsymbol{a}^{1:i-1}, a^i)\right) da^i \tag{26}$$

$$= \prod_{i=1}^{n} Z^i(s). \tag{27}$$

Beyond the general case, if we consider a special case that the old policy $\boldsymbol{\mu}$ is in a parallel manner (i.e., $\boldsymbol{\mu} = \prod_{i=1}^{n} \mu^i(a^i|s)$) while the target policy remains in a sequential manner, we can still derive similar results, differing only by the modification from $\mu^i(a^i|s, \boldsymbol{a}^{1:i-1})$ to $\mu^i(a^i|s)$.

### A.3 PROOF OF MONOTONIC IMPROVEMENT WITH BINARY FILTERING

**Proposition 1.** *(Relationship between Exponential Weighting and Binary Filtering). In terms of the weight $e^{A_{\boldsymbol{\mu}}^i(s, \boldsymbol{a}^{1:i-1}, a^i)/\beta}$ in Exponential Weighting where $\beta > 0$, for any $A_{\boldsymbol{\mu}}^i(s, \boldsymbol{a}^{1:i-1}, a^i) < 0$, we have the following limitation,*

$$\lim_{\beta \to 0^+} \exp\left(\frac{A_{\boldsymbol{\mu}}^i(s, \boldsymbol{a}^{1:i-1}, a^i)}{\beta}\right) = 0, \quad \text{for } \forall A_{\boldsymbol{\mu}}^i(s, \boldsymbol{a}^{1:i-1}, a^i) < 0 \tag{28}$$

*As $\beta \to 0^+$, Exponential Weighting becomes a special case – Binary Filtering where the samples with $A_{\boldsymbol{\mu}}^i(s, \boldsymbol{a}^{1:i-1}, a^i) < 0$ are filtered out.*

*Proof.* We first define the minimum of the absolute value of those negative $A_{\boldsymbol{\mu}}^i$,

$$\alpha = \min_{A_{\boldsymbol{\mu}}^i < 0} |A_{\boldsymbol{\mu}}^i| = \min_{A_{\boldsymbol{\mu}}^i < 0} -A_{\boldsymbol{\mu}}^i$$

To achieve Eq. (28), we only need to ensure that the rate at which $e^{A_{\boldsymbol{\mu}}^i(s, \boldsymbol{a}^{1:i-1}, a^i)/\beta}$ approaches zero is faster than the rate at which $\beta$ approaches zero. One way to guarantee this is to choose $\beta$ such that it is proportional to the absolute value of $A$. Thus, we define $\beta = k \cdot \alpha$ where $k$ is a positive hyperparameter. Then we have,

$$\exp\left(\frac{A_{\boldsymbol{\mu}}^i(s, \boldsymbol{a}^{1:i-1}, a^i)}{\beta}\right) \leq \exp\left(\frac{-\alpha}{\beta}\right) = \exp\left(\frac{-1}{k}\right)$$

Finally, for any positive $\epsilon > 0$, there exists a positive $k > 0$, it holds the following:

$$\exp\left(\frac{-1}{k}\right) < \epsilon$$

Taking the natural logarithm of both sides, we get:

$$k \ln(\epsilon) + 1 > 0 \tag{29}$$

With an arbitrary $\epsilon > 0$, we can always find a $k$ that satisfies Eq. (29), further satisfying Eq. (28). $\qquad \square$

**Proposition 2.** *(Policy improvement with Binary Filtering). By behaviour cloning (BC) on a filtered dataset with Binary Filtering $\mathbb{1}[A_{\boldsymbol{\mu}}^i(s, \boldsymbol{a}^{1:i-1}, a^i) > \epsilon]$ where $\epsilon \geq 0$, new policy $\boldsymbol{\pi}$ is superior to the basic policy $\boldsymbol{\mu}$, i.e., $J(\boldsymbol{\pi}) - J(\boldsymbol{\mu}) > 0$.*

*Proof.* According to BC on a filtered dataset with *Binary Filtering* $\mathbb{1}[A_{\boldsymbol{\mu}}^i(s, \boldsymbol{a}^{1:i-1}, a^i) > \epsilon]$, we have:

$$\pi^i(a^i|s, \boldsymbol{a}^{1:i-1}) = \frac{\mathbb{1}[A_{\boldsymbol{\mu}}^i(s, \boldsymbol{a}^{1:i-1}, a^i) > \epsilon]\mu^i(a^i|s, \boldsymbol{a}^{1:i-1})}{Z^i(s)} \tag{30}$$

where $Z^i(s)$ is the partition function. Given the new policy $\boldsymbol{\pi}(\boldsymbol{a}|s) = \prod_{i=1}^n \pi^i(a^i|s, \boldsymbol{a}^{1:i-1})$, the expected improvement from Eq. (6) can be rewritten as,

$$\hat{\eta}(\boldsymbol{\pi}) = \mathbb{E}_{s\sim\rho_{\boldsymbol{\mu}}(s), \boldsymbol{a}\sim\boldsymbol{\pi}(\boldsymbol{a}|s)}\left[A_{\boldsymbol{\mu}}(s, \boldsymbol{a})\right]$$
$$= \mathbb{E}_{s\sim\rho_{\boldsymbol{\mu}}(s)}\mathbb{E}_{a^1\sim\pi^1(a^1|s)}\mathbb{E}_{a^2\sim\pi^2(a^2|s, a^1)}\cdots\mathbb{E}_{a^n\sim\pi^n(a^n|s, \boldsymbol{a}^{1:n-1})}\left[A_{\boldsymbol{\mu}}(s, \boldsymbol{a})\right]$$

Substituting Lemma 1 and Eq. (30) into the above equation, we get:

$$\hat{\eta}(\boldsymbol{\pi}) = \mathbb{E}_{s\sim\rho_{\boldsymbol{\mu}}(s)}\mathbb{E}_{a^1\sim\pi^1(a^1|s)}\mathbb{E}_{a^2\sim\pi^2(a^2|s, a^1)}\cdots\mathbb{E}_{a^n\sim\pi^n(a^n|s, \boldsymbol{a}^{1:n-1})}\left[\sum_{i=1}^n A_{\boldsymbol{\mu}}^i(s, \boldsymbol{a}^{1:i-1}, a^i)\right]$$

$$= \mathbb{E}_{s\sim\rho_{\boldsymbol{\mu}}(s)}\left[\sum_{i=1}^n \mathbb{E}_{a^i\sim\pi^i(a^i|s, \boldsymbol{a}^{1:i-1})}\left(A_{\boldsymbol{\mu}}^i(s, \boldsymbol{a}^{1:i-1}, a^i)\right)\right]$$

$$= \mathbb{E}_{s\sim\rho_{\boldsymbol{\mu}}(s)}\left[\sum_{i=1}^n \mathbb{E}_{a^i\sim\mu^i(a^i|s, \boldsymbol{a}^{1:i-1})}\left(\frac{\mathbb{1}[A_{\boldsymbol{\mu}}^i(s, \boldsymbol{a}^{1:i-1}, a^i) > \epsilon]A_{\boldsymbol{\mu}}^i(s, \boldsymbol{a}^{1:i-1}, a^i)}{Z^i(s)}\right)\right] \tag{31}$$

And we note that the expected improvement from Eq. (6) entails the following relationship,

$$\hat{\eta}(\boldsymbol{\mu}) = J(\boldsymbol{\mu}) - J(\boldsymbol{\mu}) = \mathbb{E}_{s\sim\rho_{\boldsymbol{\mu}}(s), \boldsymbol{a}\sim\boldsymbol{\mu}(\boldsymbol{a}|s)}\left[A_{\boldsymbol{\mu}}(s, \boldsymbol{a})\right]$$

$$= \mathbb{E}_{s\sim\rho_{\boldsymbol{\mu}}(s)}\left[\sum_{i=1}^n \mathbb{E}_{a^i\sim\mu^i(a^i|s, \boldsymbol{a}^{1:i-1})}\left(A_{\boldsymbol{\mu}}^i(s, \boldsymbol{a}^{1:i-1}, a^i)\right)\right]$$

$$= 0 \tag{32}$$

Comparing Eq. (31) with Eq. (32), it is obvious that those local advantages $A_{\boldsymbol{\mu}}^i(s, \boldsymbol{a}^{1:i-1}, a^i)$ below the threshold $\epsilon$ would not be calculated in the expectation $\hat{\eta}(\boldsymbol{\pi})$. Hence, when the threshold $\epsilon \geq 0$ it naturally holds $\hat{\eta}(\boldsymbol{\pi}) > \hat{\eta}(\boldsymbol{\mu}) = 0$, i.e., $J(\boldsymbol{\pi}) - J(\boldsymbol{\mu}) > 0$. $\qquad \square$

## B  ADDITIONAL RELATED WORKS

**Other LLM-based Embodied Agent.**   Beyond task planning, LLMs also shoulder other roles for embodied agents. (i) **Foundation Policy.** Robot Transformer (Brohan et al., 2023b;a), PaLM-E (Driess et al., 2023), Open-X (Collaboration, 2023), and RoboFlamingo (Li et al., 2023b) use pre-trained LLM or VLM as the foundation policies and fine-tune the parameters with embodied data from real-world tasks. The LLM tokens and action tokens of agents are unified in fine-tuning. (ii) **Code Generator**. Given high-level task descriptions, LLMs can generate executable code by calling the basic control primitives (Liang et al., 2023; Vemprala et al., 2023) or low-level actions (Wang et al., 2023b) of embodied agents. VoxPoser (Huang et al., 2023) leverages the code-writing capabilities of LLMs to compose 3D value maps via VLM and adopt model-predictive control (MPC) for planning. (iii) **Reward Designer**. Text2Reward (Xie et al., 2023a), Language2Reward (Yu et al., 2023), and Eureka (Ma et al., 2023) leverage GPT-4 to produce interpretable reward codes, and allow iterative refinement with feedback. (iv) **Data Generator**. To enhance task-level generalization, GenSim (Wang et al., 2023a) adopts LLMs to propose task curriculum and novel sub-tasks to solve complex tasks. RoboGen (Wang et al., 2023c) proposes a closed-loop process to generate robot data, including proposing tasks, generating simulation environments, decomposing sub-tasks, and solving sub-tasks via RL or MPC.

## C  ALGORITHMIC DESCRIPTION

In this section, we give the algorithm descriptions of critic regression via Monte Carlo estimation, as well as the process of *ReAd-S* and *ReAd-J* algorithms. We highlight the difference between *ReAd-S* and *ReAd-J* by different colors.

---

**Algorithm 1** Critic regression on $\mathcal{D}$ following $\boldsymbol{\mu} = \boldsymbol{\pi}_{\text{llm}}$

---

**Require:** data buffer $\mathcal{D}$, batch size B, critic $Q_\theta$, the set of agents $\mathcal{N}$
**for** iteration $k = 1, ..., M$ **do**
  **for** all ordered subsets $\{i_1, i_2, ..., i_u\} \subseteq \mathcal{N}$ **do**
    compute Monte Carlo return estimates $\mathcal{R}_{s, \boldsymbol{a}^{i_{1:u}}}$

$$\mathcal{R}_{s, \boldsymbol{a}^{i_{1:u}}} = \sum_{\boldsymbol{a}^{-i_{1:u}} \in \mathcal{D}} \sum_{t=0}^{T} \gamma^t r_t$$

    update estimated critic $Q_\theta^{i_{1:u}}$ by using

$$\underset{Q_{\boldsymbol{\mu}}^{i_{1:u}}}{\arg \min} \, \mathbb{E}_{s, \boldsymbol{a}^{i_{1:u}} \sim \mathcal{D}} \left[ \left\| \mathcal{R}_{s, \boldsymbol{a}^{i_{1:u}}} - Q_{\boldsymbol{\mu}}^{i_{1:u}} \right\|^2 \right]$$

  **end for**
**end for**

---

---

**Algorithm 2** *ReAd-S*: Reinforced Advantage Feedback with Sequential Individual Plan Refinement

---

**Require:** agent name $u^1, ..., u^N$, task horizon $T$, refinement threshold $\alpha$, history buffer $H$, critic $Q_\theta$

**Denotation:** dialog $d$; agent $u^i$'s plan $a^i$

initialize timestep $t \leftarrow 0$

initialize observation $s_0 \leftarrow$ env.reset()

**while** $t < T$ **do**

    initialize joint action $\boldsymbol{a}_t = \{\}$ and history $H = \{\}$

    set $\alpha \leftarrow 2\alpha$

    **for** $i = 1, ..., N$ **do**

        initialize the history of evaluated action-score pairs $\mathcal{P} = \{\}$

        **repeat**

            $d, a_t^i \leftarrow$ LLMPrompt$(H, s_t, u_t^i, \mathcal{P})$

            $\mathbb{S}_{\text{ReAd-S}}(a_t^i) = Q_\theta^{1:i}(s_t, \boldsymbol{a}_t^{1:i-1}, a_t^i) - Q_\theta^{1:i-1}(s_t, \boldsymbol{a}_t^{1:i-1})$

            $\mathcal{P} \leftarrow \mathcal{P} \cup \{(s_t, \boldsymbol{a}_t^{1:i-1}, a_t^i, \mathbb{S}_{\text{ReAd-S}}(a_t^i))\}$

            $\alpha \leftarrow \alpha/2$

        **until** $\mathbb{S}_{\text{ReAd-S}}(a_t^i) > \alpha$

        $H \leftarrow H \cup \{d\}$

    **end for**

    $\sigma_t \leftarrow$ MotionPlanner$(o_t, \boldsymbol{a}_t)$

    $o_{t+1}, done \leftarrow$ env.step$(\sigma_t)$

    **if** $done$ is $True$ **then**

        **break**

    **end if**

**end while**

---

**Algorithm 3** *ReAd-J*: Reinforced Advantage Feedback with Joint Plan Refinement

---

**Require:** agent name $u^1, ..., u^N$, task horizon $T$, pick action threshold $\alpha$, history buffer $H$, critic $Q_\theta$, discount factor $\gamma$

**Denotation:** dialog $d$; Joint WAIT action $\boldsymbol{w}$

set $H = \{\}$

initialize timestep $t \leftarrow 0$

initialize observation $s_0 \leftarrow$ env.reset()

**while** $t < T$ **do**

    set $\alpha \leftarrow 2\alpha$

    initialize the history of evaluated action-score pairs $\mathcal{P} = \{\}$

    **repeat**

        $d, \boldsymbol{a}_t \leftarrow$ LLMPrompt$(H, s_t, [u^1, ..., u^N], \mathcal{P})$

        $\mathbb{S}_{\text{ReAd-J}}(\boldsymbol{a}_t) = Q_\theta(s_t, \boldsymbol{a}_t) - \frac{1}{\gamma}Q_\theta(s_t, \boldsymbol{w})$

        $\mathcal{P} \leftarrow \mathcal{P} \cup \{(s_t, \boldsymbol{a}_t, \mathbb{S}_{\text{ReAd-J}}(\boldsymbol{a}_t))\}$

        $\alpha \leftarrow \alpha/2$

    **until** $\mathbb{S}_{\text{ReAd-J}}(\boldsymbol{a}_t) > \alpha$

    $H \leftarrow \{d\}$

    $\sigma_t \leftarrow$ MotionPlanner$(o_t, \boldsymbol{a}_t)$

    $o_{t+1}, done \leftarrow$ env.step$(\sigma_t)$

    **if** $done$ is $True$ **then**

        **break**

    **end if**

**end while**

---

## D  ENVIRONMENT DETAILS

We use Difficult Variants of RoCoBench (*DV-RoCoBench*) adapted from RoCoBench (Mandi et al., 2023) and *Overcooked-AI* (Carroll et al., 2019) in our experiments. *DV-RoCoBench* involves three tasks: Sweep Floor, Make Sandwich and Sort Cubes. And we choose two representative scenarios – Cramped Room and Forced Coordination from *Overcooked-AI* in our experiments. In this section, we

present a comprehensive overview of the task specifications along with the difficulty modifications we have made in *DV-RoCoBench* and the scenario specifications in two scenarios of *Overcooked-AI*.

As for **DV-RoCoBench**, we directly inherit the action set and quantity of robots from RoCoBench, but design diverse task goals to introduce different difficulty levels. In original RoCoBench, the action set is not the same among different tasks.

As for **Overcooked-AI**, different scenarios share the same action space but are initialized with different kitchen layouts.

### D.1 SWEEP FLOOR

**Task Description.** In this task, the two robots are positioned on opposite sides of the table. Each robot arm equipped with a dustpan and broom must collaborate to efficiently sweep all cubes of the designated color into the dustpan. Subsequently, the robot that holds the dustpan is responsible for disposing of the collected cubes in the trash bin. In this environment, two distinct types of robots with different action sets are used.

1. UR5E robot holding a dustpan ('Alice'): can move to all cubes and can perform only three operations: MOVE, DUMP, and WAIT.

2. Franka Panda holding a broom ('Bob'): can move to all cubes and can perform only three operations: MOVE, SWEEP, and WAIT.

3. Action sets: (i) MOVE [target]: target can only be a cube. (ii) DUMP: pour all cubes in the dustpan into the trash bin. (iii) SWEEP [target]: sweep the target cube into the dustpan. (iv) WAIT.

**Difficulty Settings.** We shift the task goal from sweeping away all the cubes to sweeping away the cubes of a given color. We establish 5 distinct difficulty levels based on the number of cubes and the number of the target cubes. By increasing the difficulty level step by step, the quantity of all cubes and the cubes of a given color increase also gradually, as shown in Figure 5.

| Y1_G1 | Y1_G2 | Y2_G2 | Y2_G3 | Y3_G3 |
|---|---|---|---|---|

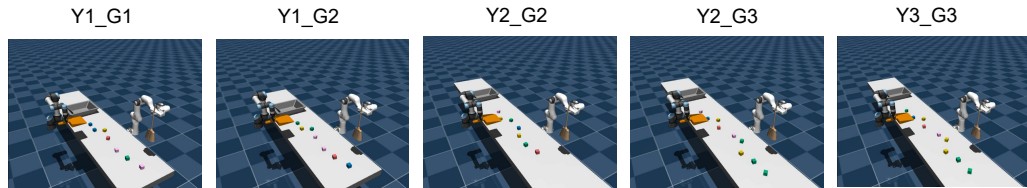

Figure 5: The initial states of the 5 difficulty levels in modified Sweep Floor. The yellow and green squares are the ones to be swept in this task. The first three tasks have a total of 7 squares, while the last two have 9. We assess task difficulty based on the number of cubes to be swept and the total cube number. For example, the Y1_G1 in the figure represents 1 yellow cube and 1 green cube needs to be swept.

### D.2 MAKE SANDWICH

**Task Description.** In this task, two robots are positioned on opposite sides of a table to assemble a sandwich based on a given recipe, requiring collaborative effort to collect and stack the ingredients in the specified order until all components have been properly arranged. This environment accommodates two distinct types of robots capable of executing all actions in the action set. Each robot has a restricted range to manipulate the cubes.

1. UR5E robot ('Chad'): can only retrieve the food on the right side.

2. Humanoid robot ('Dave'): can only retrieve the food on the left side.

3. Action set: 1) PICK [object]: object must be a food. 2) PUT [object] on [target]: object must be a food and target could be a food, cutting_board, or table. 3) WAIT.

**Difficulty Settings.** We establish 4 distinct difficulty levels dependent on the length of the recipe. A longer recipe requires more complex collaboration between humanoid and robot arm. The recipe lengths for these different settings are set to 3, 5, 7, and 9, respectively, as shown in Figure 6.

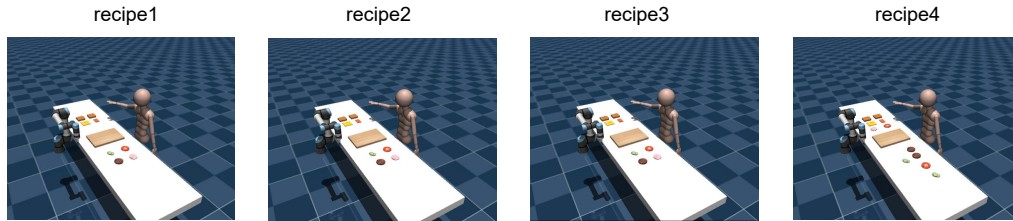

Figure 6: The initial states of the 4 difficulty levels in modified Make Sandwich. The initial three tasks shared the same food and layout, differing only in the length of the recipe. Conversely, the final task presented distinct food and layout, accompanied by a lengthier recipe. The recipe lengths for four tasks are set to 3, 5, 7, and 9, respectively.

### D.3 SORT CUBES

**Task Description.** The task requires three robots positioned on opposite sides of a table to collaboratively place three target blocks in specific locations, utilizing their limited range of motion and assisting each other as needed. The current environment consists of three robots capable of executing all actions in the action set, albeit with limited mobility range.

1. UR5E with robotic gripper ('Alice'): must put the blue square on panel2, can only reach: panel1, panel2, panel3.
2. Franka Panda ('Bob'): must put pink polygon on panel4, can only reach: panel3, panel4, panel5.
3. UR5E with suction gripper ('Chad'): must put yellow trapezoid on panel6, can only reach: panel5, panel6, panel7.
4. Action set: 1) PICK [object] PLACE [panelX]: the object must be a cube and panelX cannot be the target panel of another cube. 2) WAIT.

**Difficulty Settings.** We establish 5 difficulty levels based on the distance of the three blocks towards their corresponding target location. Since each robot has limited range of motion, picking further cube to the target location requires more complex collaboration between three robot arms.

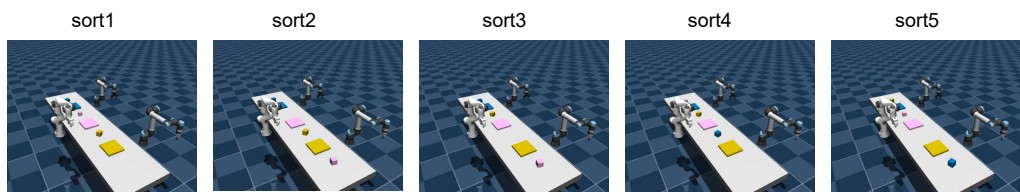

Figure 7: The initial states of the 5 difficulty levels in modified Sort Cubes. In these tasks, we orchestrated the initial placement of each block, and gauged difficulty based on the cumulative distance between the three blocks and the target panel. The shape of the three cubes was modified to avoid the robot's inability to pick up the objects due to their shape.

### D.4 OVERCOOKED-AI

In *Overcooked-AI*, two agents are originally required to make as much soup as possible in limited timesteps with high coordination efficiency. Agents place a specified number of onions in a pot, leave them to cook for a specified number of timesteps, put the resulting soup in a dish, and serve it, giving

all agents a reward. The capacity of all agents to pick up items is 1. Every agent can only carry 1 item such as the dish and the onion. In our experiment, to enable measuring with the success rate metric, we modify the task as cooking and delivering a soup to the service counter within a specified number of timesteps. The action set of this environment are as following:

1. north: agent moves one step north. If agent collides with another object, it will not move.

2. south: agent moves one step south. Same as the previous term.

3. east: agent moves one step east. Same as the previous term.

4. west: agent moves one step west. Same as the previous term.

5. interact: agent interacts with a object, including picking up or putting down an item, turning on the cooking table, and putting the cooked soup in the dish.

6. stay: agent does nothing.

The first four actions (north, south, east and west) cover the movement of the agent, and the interact action enables the interaction between the agent and other objects. We use Figure 8 to explain the above rules:

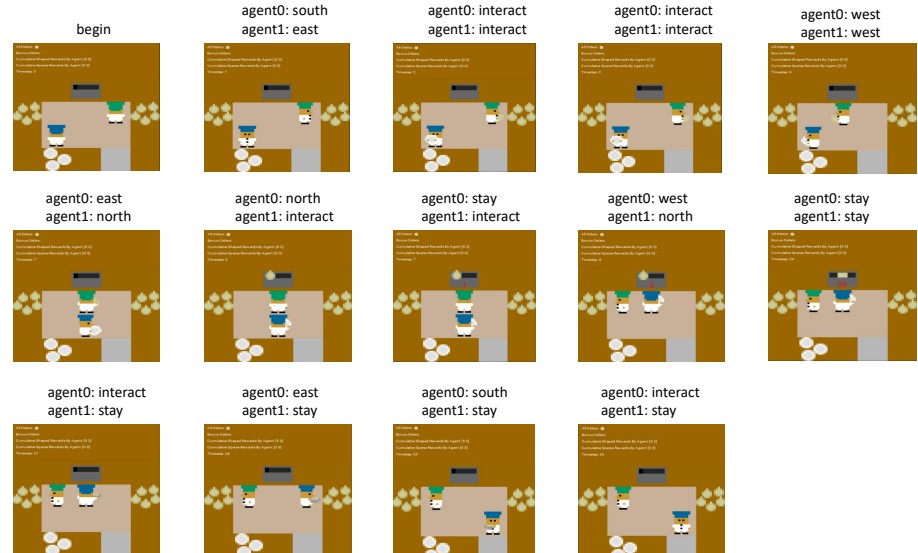

Figure 8: In 2nd frame, since both agents collide with the workbench, the agents merely change their current orientation. In 4th frame, since both agents have picked up an object in their hands, executing "interact" again will not pick up additional items. In 7th frame, agent1 places the onion on the cooking table. And in 8th frame, agent1 turns on the cooking table and starts cooking. In 10th and 11th frames, the soup is done and then put in a dish by agent0. In the last frame, agent0 serves the cooked soup.

**Cramped Room.**    Two agents collaborate in a relatively small kitchen, and thus two agents must be extremely careful to avoid collisions in order to complete the cooking task as quickly as possible. The scenario is shown in the Figure 8.

**Forced Coordination.**    The working spaces of two agents are completely separated, where one agent only has access to the cooking table and the service counter and the other only has access to onions and dishes. The scenario is shown in the Figure 9.

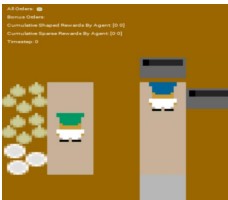

Figure 9: In this task, agent0 must wait for agent1 to deliver the onion to the table before agent0 can place it on the cooking table, and after the soup is ready, agent0 must wait for agent1 to place the plate on the table before it can serve the soup and deliver it to the service table.

# E  ADDITIONAL EXPERIMENTAL RESULTS

In this section, we give the detailed experiment results of 3 tasks in *DV-RoCoBench* and 2 scenarios in *Overcooked-AI*. We also show the execution screenshots of our method and baselines in the representative environments.

## E.1  COMPARISON OF BASELINES

Table 3: Overview of the key properties that distinguish four methods. (i) **State Type**: whether the environment state included in the prompt is global or not; (ii) **Planning Scheme**: whether LLM output plans sequentially or not; (iii) **History Info**: whether all the history before is reserved in the prompt or not.

|  | STATE TYPE | PLANNING SCHEME | HISTORY INFO | FEEDBACK TYPE |
|---|---|---|---|---|
| RoCo | PARTIAL | SEQUENTIAL | ALL PREVIOUS ROUNDS | PHYSICAL VERIFICATION |
| ReAd-S | PARTIAL | SEQUENTIAL | LAST ROUND | ADVANTAGE SCORE |
| CENTRAL-PLAN | GLOBAL | PARALLEL | ALL PREVIOUS ROUNDS | PHYSICAL VERIFICATION |
| ReAd-J | GLOBAL | PARALLEL | LAST ROUND | ADVANTAGE SCORE |
| ReAct | GLOBAL | PARALLEL | ALL PREVIOUS ROUNDS | PHYSICAL VERIFICATION |
| REFLEXION | GLOBAL | PARALLEL | ALL PREVIOUS ROUNDS | PHYSICAL VERIFICATION |
| MINDAGENT | GLOBAL | PARALLEL | ALL PREVIOUS ROUNDS | PHYSICAL VERIFICATION |

## E.2  MAIN EXPERIMENTS

The results of all experiments are shown in Table 4, and Table 5. SR, NQ and ES represent success rates, the average number of requests to LLMs, and rounds of environment interactions, respectively. We have provided a detailed introduction to these metrics in §5.1.

## E.3  EXTENDED EXPERIMENT WITH LLAMA-3.1-70B-INSTRUCT

Here, we instead use Llama-3.1-70B-Instruct (Dubey et al., 2024) as the basic LLM policy to validate that our algorithm can improve the performance of not only the closed-source models but also the open-source models. We select *Y2_G3* as the task for evaluation, and compare our *ReAd-J* with other baselines including Central Plan, ReAct, Reflexion and MindAgent. The result is reported in Table 6. In terms of the prompt and generation parameters of Llama 3.1-70B in additional experiments, we keep the prompt essentially unchanged. We coarsely search for suitable parameters for the Llama 3.1 70B instruct model. The current generation parameters are determined by a simple grid search on them. Finally, we set the temperature as 0 and $\text{top}_p$ as 0.1.

Most methods have a 10%-20% decline in SR, with a slight increase in NQ and ES. Judging from the performance of task *Y2_G3*, GPT-4 has better performance than the Llama-3.1-70B-Instruct. Although using an open-source model like Llama 3.1-70B might result in suboptimal performance, our *ReAd-J* significantly outperforms other baselines based on the same LLM, demonstrating the effectiveness of our method.

## E.4  VISUALIZATION OF ROBUSTNESS EVALUATION

We visualize the robustness comparison between *ReAd-S* and *RoCo* for accomplishing *Make Sandwich* recipe3 task when the environment resets at timestep $n = 2$, as shown in Figure 10 and Figure 11.

Table 4: The detailed results of the comparison in different tasks with various difficulty levels in *DV-RoCoBench*. The mean value and standard error are calculated over 10 random seeds.

| | | RoCo | | | ReAct | | | Central Plan | | | Reflexion | | |
|---|---|---|---|---|---|---|---|---|---|---|---|---|---|
| | | SR | NQ | ES | SR | NQ | ES | SR | NQ | ES | SR | NQ | ES |
| SWEEP | Y1_G1 | 0.9±0.32 | 14.4±5.95 | 6.2±3.12 | 1.0±0.00 | 5.5±0.50 | 5.5±0.50 | 0.4±0.52 | 15.3±0.48 | 11.2±4.92 | 1.0±0.00 | 5.0±0.00 | 5.0±0.00 |
| | Y1_G2 | 1.0±0.00 | 24.2±4.18 | 8.9±1.45 | 1.0±0.00 | 8.2±0.25 | 8.2±0.25 | 1.0±0.00 | 7.8±1.99 | 7.8±1.99 | 1.0±0.00 | 7.0±0.00 | 7.0±0.00 |
| | Y2_G2 | 1.0±0.00 | 29.1±5.40 | 10.6±1.35 | 1.0±0.00 | 10.0±0.00 | 10.0±0.00 | 0.8±0.42 | 12.7±1.77 | 12.7±1.77 | 1.0±0.00 | 10.1±0.10 | 10.0±0.00 |
| | Y2_G3 | 0.7±0.48 | 36.7±6.63 | 13.5±1.27 | 0.6±0.16 | 14.4±0.67 | 13.8±0.33 | 0.2±0.42 | 14.6±0.97 | 14.6±0.97 | 0.7±0.15 | 14.3±0.87 | 12.9±0.48 |
| | Y3_G3 | 0.6±0.52 | 41.8±7.73 | 14.7±0.48 | 0.4±0.16 | 15.2±0.25 | 14.9±0.32 | 0.0±0.00 | 15.0±0.00 | 15.0±0.00 | 0.3±0.15 | 15.1±0.23 | 14.9±0.10 |
| SANDWICH | RECIPE1 | 1.0±0.00 | 13.2±3.74 | 4.7±0.67 | 1.0±0.00 | 4.0±0.00 | 4.0±0.00 | 1.0±0.00 | 6.2±0.63 | 4.0±0.00 | 1.0±0.00 | 5.0±0.00 | 4.0±0.00 |
| | RECIPE2 | 0.9±0.32 | 28.9±11.25 | 9.1±2.42 | 1.0±0.00 | 6.0±0.00 | 6.0±0.00 | 1.0±0.00 | 8.2±0.42 | 6.0±0.00 | 1.0±0.00 | 6.8±0.13 | 6.0±0.00 |
| | RECIPE3 | 0.8±0.42 | 33.7±10.00 | 11.5±2.99 | 0.7±0.15 | 12.9±2.61 | 10.1±1.07 | 1.0±0.00 | 10.2±0.42 | 8.0±0.00 | 0.6±0.16 | 14.9±2.47 | 10.8±1.14 |
| | RECIPE4 | 0.5±0.53 | 43.1±17.84 | 13.1±2.47 | 0.6±0.16 | 16.7±2.60 | 12.5±0.75 | 0.4±0.52 | 80.5±53.35 | 14.2±1.14 | 0.5±0.17 | 17.7±2.39 | 13.1±0.67 |
| SORT | SORT1 | 1.0±0.00 | 3.3±0.95 | 1.1±0.32 | 1.0±0.00 | 1.2±0.13 | 1.0±0.00 | 1.0±0.00 | 1.0±0.00 | 1.0±0.00 | 1.0±0.00 | 1.2±0.13 | 1.0±0.00 |
| | SORT2 | 1.0±0.00 | 13.5±4.67 | 3.4±0.52 | 0.6±0.16 | 14.8±4.56 | 7.8±1.96 | 1.0±0.00 | 16.9±9.13 | 2.6±0.52 | 1.0±0.00 | 5.5±0.48 | 2.9±0.10 |
| | SORT3 | 1.0±0.00 | 18.6±15.10 | 4.9±2.60 | 0.8±0.13 | 19.4±6.18 | 6.4±1.45 | 1.0±0.00 | 8.3±4.32 | 2.3±0.95 | 1.0±0.00 | 6.6±0.50 | 4.7±0.33 |
| | SORT4 | 1.0±0.00 | 24.8±9.37 | 6.4±1.78 | 0.8±0.13 | 24.0±11.31 | 6.1±1.49 | 1.0±0.00 | 37.2±25.05 | 7.1±2.77 | 0.7±0.13 | 19.2±6.83 | 7.1±1.45 |
| | SORT5 | 1.0±0.00 | 38.5±9.96 | 7.4±2.95 | 0.7±0.15 | 17.3±3.00 | 8.4±1.59 | 0.6±0.52 | 128.4±115.99 | 11.0±3.97 | 0.8±0.13 | 13.9±3.97 | 6.9±1.43 |
| AVERAGE | | **0.89±0.19** | 25.99±8.06 | 8.25±1.74 | 0.80±0.09 | 12.11±2.29 | 8.19±0.69 | 0.74±0.17 | 25.88±15.32 | 8.39±1.36 | 0.83±0.06 | 10.16±1.24 | 7.59±0.41 |

| | | MIND | | | READ-S | | | READ-J | | |
|---|---|---|---|---|---|---|---|---|---|---|
| | | SR | NQ | ES | SR | NQ | ES | SR | NQ | ES |
| SWEEP | Y1_G1 | 1.0±0.00 | 5.0±0.00 | 5.0±0.00 | 1.0±0.00 | 10.4±0.52 | 5.0±0.00 | 1.0±0.00 | 5.9±0.99 | 5.0±0.00 |
| | Y1_G2 | 1.0±0.00 | 7.1±0.10 | 7.1±0.10 | 1.0±0.00 | 14.4±0.84 | 7.0±0.00 | 1.0±0.00 | 7.6±0.70 | 7.0±0.00 |
| | Y2_G2 | 1.0±0.00 | 9.9±0.18 | 9.8±0.13 | 1.0±0.00 | 19.9±3.28 | 9.4±0.70 | 1.0±0.00 | 13.0±4.32 | 9.0±0.00 |
| | Y2_G3 | 0.7±0.15 | 13.4±0.48 | 13.4±0.48 | 0.9±0.32 | 26.8±5.20 | 12.2±1.32 | 1.0±0.00 | 16.4±6.02 | 11.7±1.49 |
| | Y3_G3 | 0.2±0.13 | 15.1±0.10 | 15.0±0.00 | 0.8±0.42 | 31.4±3.50 | 14.0±0.82 | 0.8±0.42 | 16.4±1.71 | 13.4±0.84 |
| SANDWICH | RECIPE1 | 1.0±0.00 | 5.1±0.10 | 4.0±0.00 | 1.0±0.00 | 10.5±4.74 | 4.2±0.42 | 1.0±0.00 | 4.3±0.48 | 4.0±0.00 |
| | RECIPE2 | 1.0±0.00 | 6.6±0.16 | 6.0±0.00 | 1.0±0.00 | 14.5±2.46 | 6.4±0.52 | 1.0±0.00 | 6.5±0.85 | 6.0±0.00 |
| | RECIPE3 | 0.7±0.16 | 12.4±1.92 | 10.1±1.07 | 1.0±0.00 | 22.1±5.22 | 8.9±0.88 | 1.0±0.00 | 14.6±8.04 | 8.9±1.00 |
| | RECIPE4 | 0.6±0.16 | 16.5±2.24 | 12.7±0.72 | 1.0±0.00 | 27.9±8.06 | 11.1±1.73 | 1.0±0.00 | 10.8±0.42 | 10.0±0.00 |
| SORT | SORT1 | 1.0±0.00 | 1.2±0.13 | 1.0±0.00 | 1.0±0.00 | 3.4±0.52 | 1.0±0.00 | 1.0±0.00 | 1.1±0.32 | 1.1±0.32 |
| | SORT2 | 1.0±0.00 | 6.1±1.12 | 3.2±0.33 | 1.0±0.00 | 10.8±2.53 | 3.1±0.32 | 1.0±0.00 | 7.3±2.91 | 3.3±0.48 |
| | SORT3 | 0.8±0.13 | 11.1±3.70 | 6.2±1.54 | 1.0±0.00 | 17.5±2.80 | 3.9±0.57 | 1.0±0.00 | 8.3±3.80 | 3.4±0.84 |
| | SORT4 | 0.9±0.10 | 22.6±9.62 | 5.9±1.12 | 1.0±0.00 | 21.6±7.07 | 3.7±0.67 | 1.0±0.00 | 18.8±6.29 | 4.3±0.95 |
| | SORT5 | 0.8±0.13 | 18.0±4.12 | 7.8±1.35 | 1.0±0.00 | 33.5±6.35 | 6.1±0.88 | 1.0±0.00 | 17.3±11.87 | 4.4±1.26 |
| AVERAGE | | 0.84±0.07 | 10.72±1.71 | 7.66±0.49 | **0.98±0.05** | 18.91±3.79 | 6.86±0.63 | **0.99±0.03** | 10.59±3.48 | 6.54±0.51 |

Table 5: The detailed results of the comparison in two scenarios in *Overcooked-AI*. The mean value and standard error are calculated over 10 random seeds.

| | CRAMPED_ROOM | | | FORCED_COORDINATION | | | AVERAGE | | |
|---|---|---|---|---|---|---|---|---|---|
| | SR | NQ | ES | SR | NQ | ES | SR | NQ | ES |
| REACT | 0.0±0.00 | 20.1±0.10 | 20.0±0.00 | 0.0±0.00 | 26.9±0.75 | 25.0±0.00 | 0.00±0.00 | 23.50±0.43 | 22.50±0.00 |
| REFLEXION | 0.0±0.00 | 20.0±0.00 | 20.0±0.00 | 0.0±0.00 | 26.1±0.60 | 25.0±0.00 | 0.00±0.00 | 23.05±0.30 | 22.50±0.00 |
| MINDAGENT | 0.0±0.00 | 20.8±0.47 | 20.0±0.00 | 0.0±0.00 | 26.9±0.80 | 25.0±0.00 | 0.00±0.00 | 23.85±0.64 | 22.50±0.00 |
| CENTRAL | 0.0±0.00 | 20.0±0.00 | 20.0±0.00 | 0.0±0.00 | 25.0±0.00 | 25.0±0.00 | 0.00±0.00 | 22.50±0.00 | 22.50±0.00 |
| READ-J | **0.4±0.16** | 23.9±1.49 | 18.9±0.59 | **0.3±0.15** | 27.2±0.53 | 24.8±0.20 | **0.35±0.16** | 25.55±1.01 | 21.85±0.40 |

Table 6: The detailed result of the comparison in the task *Y2_G3* with Llama-3.1-70B-Instruct as the basic LLM.

| | ReAd-J | Central Plan | ReAct | Reflexion | MindAgent |
|---|---|---|---|---|---|
| SR | 0.9±0.10 | 0.0±0.00 | 0.4±0.16 | 0.5±0.17 | 0.7±0.15 |
| NQ | 13.6±0.56 | 15.0±0.00 | 15.0±0.00 | 13.7±0.37 | 14.3±0.15 |
| ES | 11.8±0.42 | 15.0±0.00 | 15.0±0.00 | 13.6±0.43 | 14.3±0.15 |

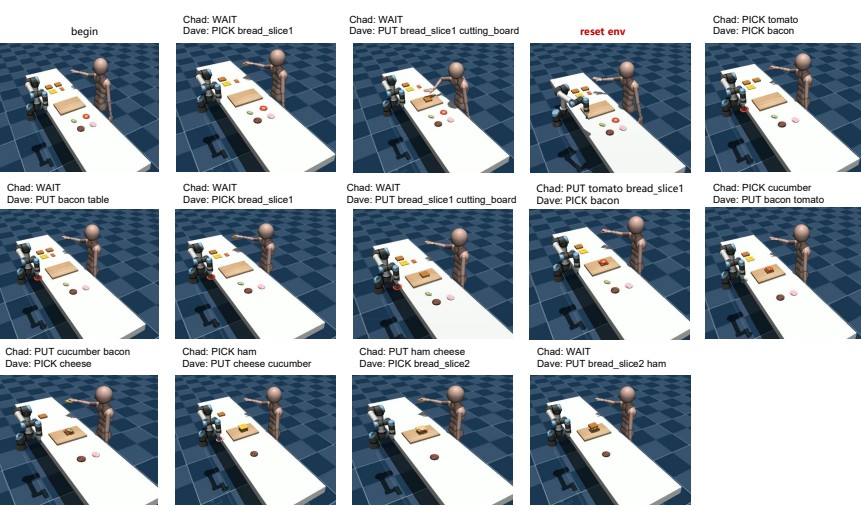

Figure 10: Screenshots of ReAd-S completing the recipe3 task in robustness test. After the environment is reset, our method will be affected by the historical dialogue information in a short period. After being prompted by the advantage function re-evaluated in the new state, our method can make a rapid re-plan based on the new state.

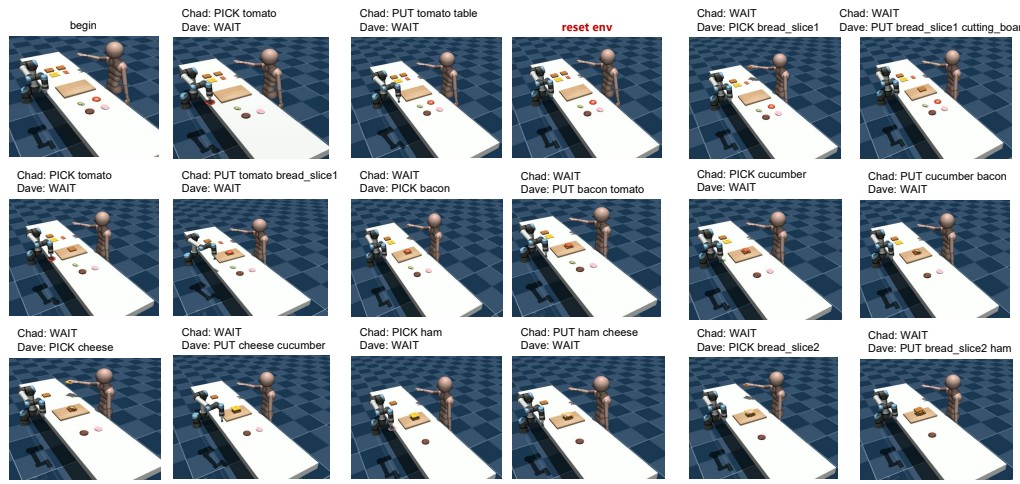

Figure 11: Screenshots of RoCo completing the recipe3 task in robustness test. RoCo needs more steps to recover from the environmental disturbance. Since the reset information is not included in the history, RoCo will be misled by historical information and require multi-round physical feedback to adjust the plan.

### E.5 DATASET AND CRITIC NETWORK

**Dataset Collection Details.** The advantage function relies on the Monte-Carlo estimation of value function with access to an offline dataset collected by $\pi_{\text{llm}}$. In practice, we employ two techniques to enhance the quality of the collected dataset. (i) We perform data collection using an LLM planner with physical verification, inspired by the RoCo policy (Mandi et al., 2023), which ensures the acquisition of high-quality interaction samples. (ii) Additionally, to address the limited state coverage issue that may arise from directly rolling out the $\pi_{\text{llm}}$ policy, we intentionally reset the environment state to an unreachable state and initiate LLM-planning from that point.

Given that our theoretical analysis demonstrates that our method can achieve a superior policy compared to the behavior policy $\mu$ through advantage-weighted regression, it is natural to consider whether a better behavior policy than $\pi_{\text{llm}}$ can be utilized for dataset collection, potentially leading to further policy improvement during optimization. Subsequently, we conduct an ablation study utilizing a mixed dataset collected by an *expert policy* and an *LLM policy*. Our preliminary findings indicate that the inclusion of additional optimal data does not result in performance improvement. We hypothesize that two reasons contribute to these unexpected results. (i) The incorporation of data from a different policy introduces increased variance in Monte-Carlo estimation, thereby reducing the stability of the value functions. Consequently, the value function may produce high-variance outputs, potentially leading to misleading optimization of the LLM planner as prompts. (ii) The LLM planner equipped with enhanced augmentation techniques achieves improved data coverage of the resulting policy. In contrast, the optimal policy is more deterministic, leading to more limited state coverage, which poses challenges for value estimation of out-of-distribution (OOD) states and actions in LLM planning. This issue bears resemblance to the distribution shift problem encountered in offline RL (Levine et al., 2020; Xie et al., 2021).

We describe the differences between *expert policy* and an *LLM policy* in detail here.

- **LLM policy**: This policy is to leverage the reasoning power of LLM to solve specific tasks and use *physical verification* as feedback. It is recommended to use a variant of *ReAd-J* for data collection, which replaces *ReAd* feedback with *physical verification* and uses only the previous round of historical information in the prompts. At each time step $t$, environment state $s_t$, robot optional actions, and task goals are added into the prompt in the form of text. And then the LLM takes the prompt as input, generates the joint action $\boldsymbol{a}_t$ of all robots and get a reward $r_t$. We store every transition as a tuple $(s_t, \boldsymbol{a}_t, r_t)$ until the task is accomplished.
- **Expert policy**: Here we implement this policy with human control. This requires a human player to analyze the task and infer the optimal action at each time step. The collected data format is the same as the method described above.

Table 7: An ablation study of data ratio of optimal data and LLM planner data in the offline dataset. The mixing ratio is represented by $\mathbf{X}\% : \mathbf{Y}\%$, where $\mathbf{X}\%$ denotes the percent of samples collected by the *LLM policy*, and $\mathbf{Y}\%$ denotes the percent of samples collected by the *optimal policy*.

|  | NQ | ES | SR |
|---|---|---|---|
| READ-J(0%:100%) | 16.4±0.54 | 13.4±0.27 | 0.8±0.13 |
| READ-J(50%:50%) | 15.8±1.12 | 13.9±0.35 | 0.6±0.16 |
| READ-J(100%:0%) | 17.6±1.89 | 13.9±0.41 | 0.7±0.15 |
| READ-S(0%:100%) | 31.4±1.11 | 14.0±0.26 | 0.8±0.13 |
| READ-S(50%:50%) | 29.1±0.91 | 13.9±0.31 | 0.7±0.15 |
| READ-S(100%:0%) | 34.2±2.18 | 14.3±0.30 | 0.5±0.17 |

**Critic Architecture.** The critic learns to estimate the value function of state-action pairs from the dataset. The state includes the environment state and the agent state, where the environment state contains variables of the simulator and the agent state is described by language. The action is also described by language. We adopt the pre-trained BERT Transformer model to extract language features of the agent state and actions. Then we concatenate the output feature with environment state features to some MLP layers to predict the $Q$-value. The structure of the critic network is given in Figure 12, and the hyper-parameters are given in Table 8.

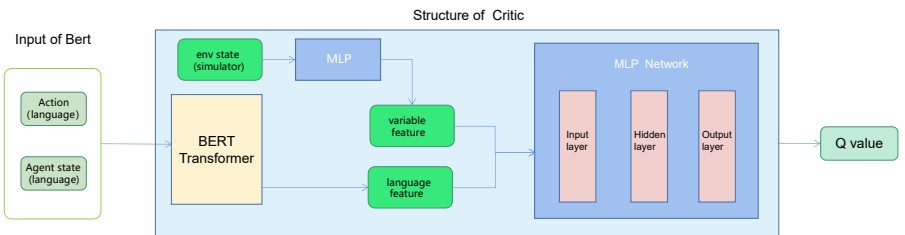

Figure 12: In this figure, the parameters of BERT Transformer are fixed and will not be updated during the training of Critic.

Table 8: The input dimensions for Critic of ReAd-J and ReAd-S are represented by JIS and SIS respectively, while HS represents the hidden layer input dimension, HN represents the number of hidden layers, LR is the learning rate, BS is batch size, TN represents the number of training iterations, SS is the dimension of environment state, and $n$ is the number of robots in the environment.

|  | JIS | SIS | HS | HN | LR | BS | TN |
|---|---|---|---|---|---|---|---|
| VALUE | 768+SS | $n \times 768$+SS | 256 | 1 | $10^{-3}$ | 32 | $9 \times 10^5$ |

**Token Consumption.** We report the details of token consumption on both benchmarks in Table 9 and Table 10 respectively. The total number of tokens consumed includes tokens consumed during pre-sampling data for training critic network. We utilize *LLM policy* to collect data for critic training in the experiment of *DV-RoCoBench*, while the data is collected by *expert policy* in the experiment of *Overcooked-AI*. Obviously, during the phase of planning, *ReAd-S* and *ReAd-J* consume less tokens than all other baselines. In terms of total consumed tokens, *ReAd-J* is comparable to the baselines which also generate joint plans in a parallel manner, and *ReAd-S* is significantly superior to RoCo.

**Critic Training.** The quantity of trajectories required for critic training depends on how challenging the task is. For 5 difficulty levels in *Sweep Floor*, critic training demands about 70, 120, 240, 600, and 1400 trajectories respectively. For 4 difficulty levels in *Make Sandwich*, about 60 trajectories are needed for critic training. For 5 difficulty levels in *Sort Cube*, critic training demands about 230, 240, 300, 400 and 510 trajectories respectively. For *Cramped room* and *Forced coordination*, the number is about 128 and 2048 respectively. It is important to note that the volume of data utilized for critic training can be adjusted flexibly to align with the specific demands and challenges of the actual situation.

Table 9: Tokens consumed by all methods during the evaluation in *DV-RoCoBench*.

| Methods | ReAd-S | ReAd-J | RoCo | Central Plan | ReAct | Reflexion | MindAgent |
|---|---|---|---|---|---|---|---|
| Tokens for planning | 9M | 6M | 24M | 15M | 11M | 11M | 13M |
| Tokens for training $\hat{Q}$ | 7M | 7M | - | - | - | - | - |
| Total tokens | 16M | 13M | 24M | 15M | 11M | 11M | 13M |

Table 10: Tokens consumed by all methods during the evaluation in *Overcooked-AI*.

| Methods | ReAd-J | Central Plan | ReAct | Reflexion | MindAgent |
|---|---|---|---|---|---|
| Tokens for planning | 1M | 2M | 4M | 3M | 4M |
| Tokens for training $\hat{Q}$ | - | - | - | - | - |
| Total tokens | 1M | 2M | 4M | 3M | 4M |

# F  EXTENDED DISCUSSION ABOUT SYMBOL GROUNDING

In this section, we would like to discuss the LLM grounding problem in embodied tasks beyond our algorithm. Currently, most of available embodied multi-agent collaboration benchmarks (e.g., *DV-RoCoBench* and *Overcooked-AI*) establish the base for LLM grounding by transforming the state/image in the environment to the textual description. Since the LLM is not capable of perceiving the current situation in the environment via visual signals, such a transformation may be achieved by directly using specific object identifiers without visual grounding. However, it may seem to ruin the purpose of LLM grounding where the main role of language is originally to provide a vehicle for establishing common ground and resolving ambiguities. It makes the evaluation of ours and other LLM-based embodied algorithms (Ahn et al., 2022; Yao et al., 2023b; Shinn et al., 2023; Gong et al., 2023) on these benchmarks possibly overestimated on solving the symbol grounding problem (Harnad, 1990).

We acknowledge that directly using fictional object identifiers without visual grounding is a limitation while at the same time it implies that a potential solution to overcome this limitation is to use strong Visual Language Models (VLMs), e.g., GPT-4o. Specifically, it requires identifying the object types (in **Make Sandwich**) or positions (in **Sort Cubes** and **Sweep Floor**), and summarizing the information with a corresponding textual representation, which aligns well with the purpose of symbol grounding. Inspired by this, we conduct a simple but essential experiment to investigate how well GPT-4o captures and describes the necessary information compared with that generated by the object identifiers. Taking the **Forced Coordination** as the test scenario, we give a example in the prompt, which includes a image of current situation of the environment paired with a textual description previously given by the human about this image. Then we ask GPT-4o for generating an appropriate response for the input image, following the template in the example. The example case and test case are shown as Figure 13, and the output textual state and ground truth textual state are listed as follows.

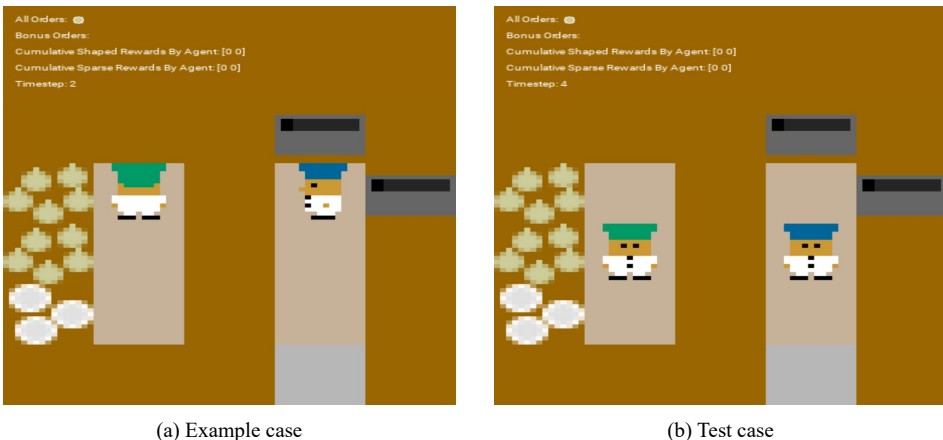

(a) Example case                                      (b) Test case

Figure 13: The example case and test case for testing the visual understanding and summarizing capability of GPT-4o.

[Prompt]:
You need to accomplish a task where you need to precisely summarize the necessary information from a given image. We start by introducing the meaning of each character appeared in the *Current Env State* which would be introduced in the example we provide later.
*Character Meaning*:
The letter X stands for table, P for cooking station, O and o stand for onion, D and d for plates, and S for service desk. When the onion or dish is on the table or being held by agent, an o or d will be added after its corresponding character. When the onion is placed on the cooking table, it will be denoted as p{ø, p{øø means that there are two onions on the cooking table. And when the cooking table cooks the soup, it will show how long it has been cooked, such as p{ø20 means that it has been cooked in 20 time steps. The numbers 1 and 0 represent the agent, and the direction arrow ↑↓←→ represents the direction the agent is facing.Each object occupies a grid size, and the agent moves one grid distance at a time.
Given a image we input, you should first describe the environment status of the image in *Current Env State*, then summarize the events according to *Character Meaning*. Now we provide a specific example. Taking the attached image as input, the standard response corresponding to this image is as follows:
*Current Env state*:
X X X P X
O ↑1 X ←0 P
O    X    X
D    X    X
X X X S X
Cook station local (0, 3) (1, 4), there are 0 onions on it, soup has not yet begun to cook
Onions local: (1, 0) (2, 0)
Dishes local: (3, 0)
Server desk local: (4, 3)
Agent0 local: (1, 3), direction: west, hold: nothing
Agent1 local: (1, 1), direction: north, hold: nothing
The above all is the instruction for the task.
[Inputting the test image observation]
[Prompt]:
Please describe this picture following the previous instruction.
[Output]:
Based on the new environment state:
*Current Env state*:
X X X P X
O ↑1 X ↑0 P
O    X    X
D    X    X
X X X S X
Cook station local: (0, 3) (1, 4), there are 0 onions on it, soup has not yet begun to cook.
Onions local: (1, 0), (2, 0).
Dishes local: (3, 0).
Server desk local: (4, 3).
Agent0 local: (1, 3), direction: north, hold: nothing.
Agent1 local: (1, 1), direction: north, hold: nothing.
This description follows the format used earlier to capture the environment's status at the current timestep.
[Ground truth]:
*Current Env state*:
X X X P X
O X P
O ↓1 X ↓0 X
D X X
X X X S X
Cook station local (0, 3) (1, 4), there are 0 onions on it, soup has not yet begun to cook
Onions local: (1, 0) (2, 0)
Dishes local: (3, 0)
Server desk local: (4, 3)
Agent0 local: (2, 3), direction: south, hold: nothing
Agent1 local: (2, 1), direction: south, hold: nothing

Shown in the above response, GPT-4o can generate a textual state with the correct format based on the image and template, but the coordinates and relative positions of objects are inconsistent with the actual situation, which has also been discussed in previous works (Xu et al., 2023). But surprisingly, it can correctly summarize the location and status of all entities in the wrong text-format array. Overall result shows that VLMs are hard to understand spatial relationship from images currently.

# G ILLUSTRATION OF THE INTERACTION PROCESS

we illustrate the distinctions between ReAd-S and RoCo by presenting a series of task execution screenshots. In Figure 14 and Figure 15, we compare the screenshots of our method and RoCo algorithm in task *Sweep Floor* Y2_G2. Our method can perform re-plan and correct the initial planning using advantage feedback, which results in a minimum number of environmental interactions. In contrast, RoCo which relies on physical feedback requires more negotiation and interactions with the environment. A similar comparison is shown in Figure 16 and Figure 17 for *Sort Cubes* sort4. A comparison between *ReAd-J* and Central Plan on *Forced Coordination* scenario is shown in Figure 18 and Figure 19.

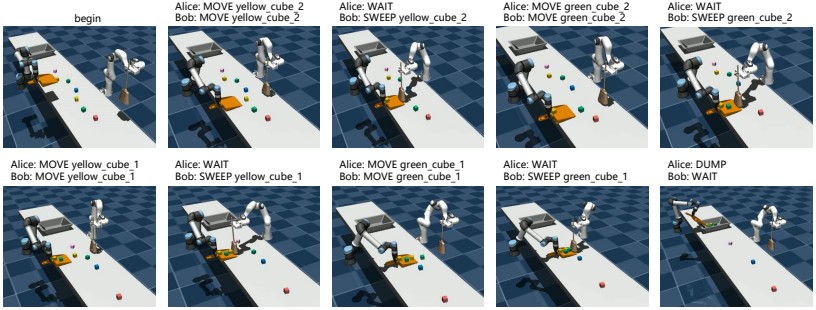

Figure 14: Snapshots of the interaction process of *ReAd-J* in task *Sweep Floor* Y2_G2. Our method obtains the minimum number of environmental interactions needed to complete the task.

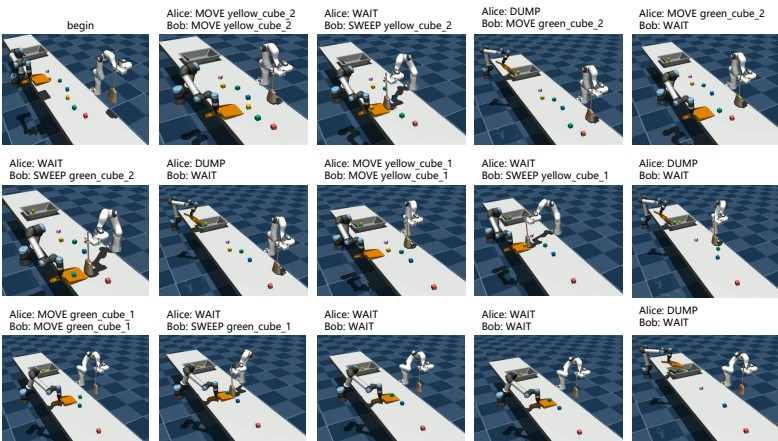

Figure 15: Snapshots of the interaction process of *RoCo* in task *Sweep Floor* Y2_G2. The figure above shows that after planning and sweeping a cube into the dustpan, RoCo will dump it into the trash bin. However, after sweeping the last cube into the dustpan, instead of immediately planning to dump it to complete the task, LLM stubbornly believes that the task is done and plans to wait for the next two interactions.

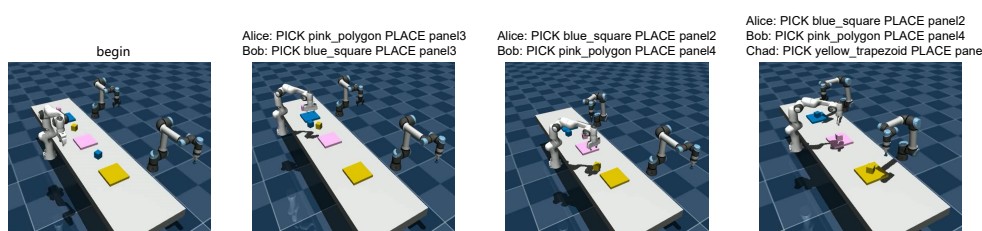

Figure 16: Snapshots of the interaction process of *ReAd-S* in task *Sort Cubes* sort4. This task is challenging and requires the collaboration of three robots and takes a minimum of three steps to complete. Our approach efficiently accomplishes this task with minimal environment interactions.

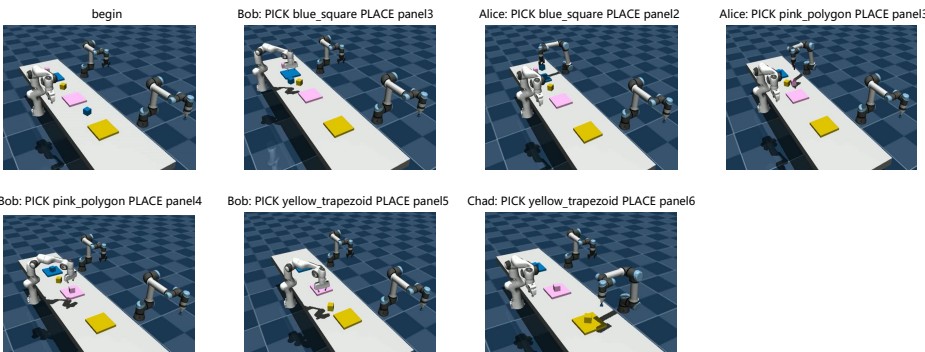

Figure 17: Snapshots of the interaction process of *RoCo* in task *Sort Cubes* sort4. Before the joint actions of all robots are executed, the planning result can only be improved through the dialogue of LLMs. In addition, environmental feedback can be generated only after the agent interacts with the simulator. In contrast, our advantage feedback provides timely feedback in the process of LLM planning for policy improvement before interaction.

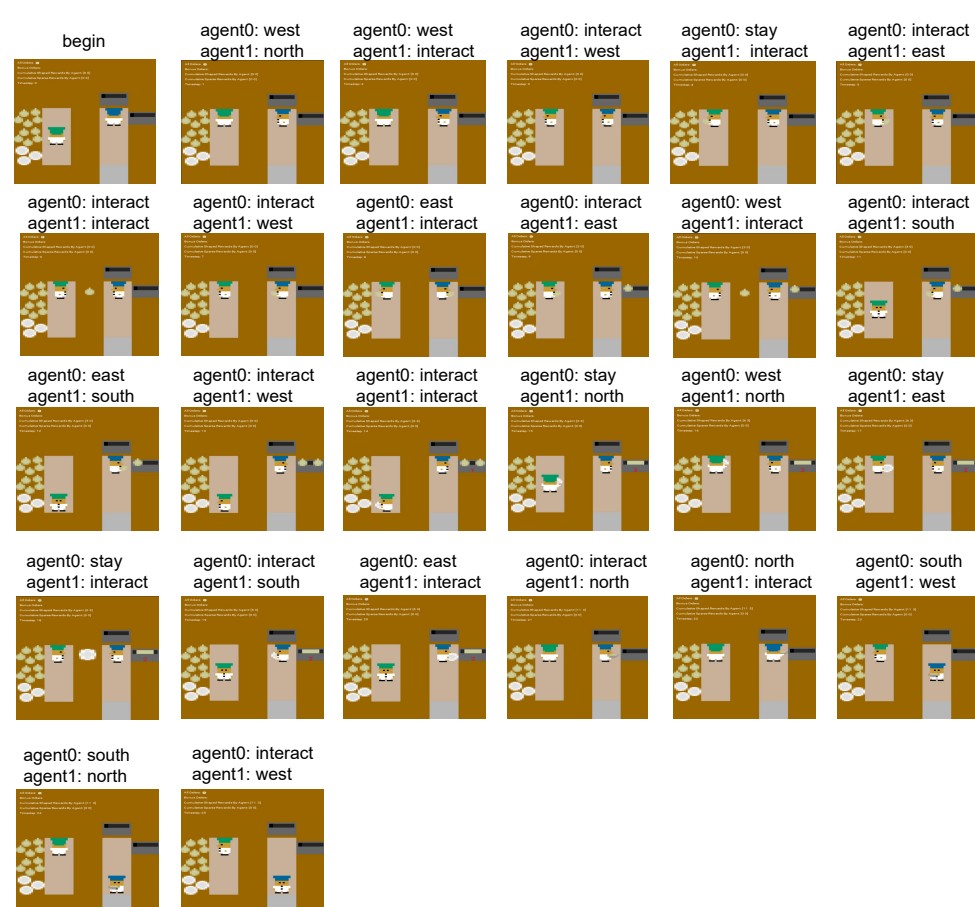

Figure 18: Snapshots of the interaction process of *ReAd-J* in task *Forced Coordination*. This task is challenging and requires the collaboration of two agents and takes a minimum of 22 steps to complete. Most of the time, ReAd can improve the unreasonable planning result generated by LLM, so that *ReAd-J* can complete the task smoothly. However, due to the out-of-distribution (OOD), it is possible to evaluate the advantage value of some unreasonable planning to carry out environmental interaction

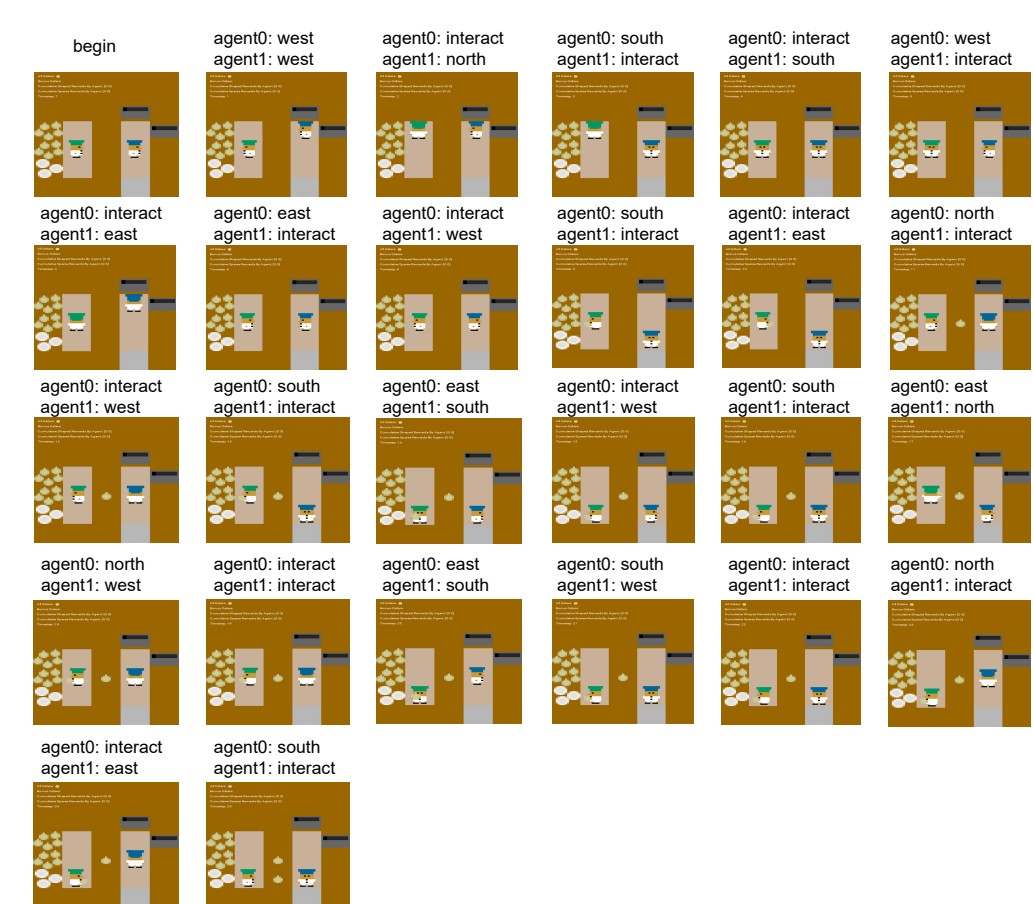

Figure 19: Snapshots of the interaction process of *Central Plan* in task *Forced Coordination*. From the screenshot of the interaction process, it can be found that in the *Forced Coordination*, it is difficult for LLM to understand the state of the current environment, so hallucination occurs frequently, resulting in the failure of *Central Plan* to effectively complete the task.

## H  Examples of Reinforced Advantage Feedback

**Format of Advantage Feedback**. The Advantage Feedback comprises of [Evaluation Score] and [Improvement Feedback]. If the advantage score of the action is higher than a threshold, the response will only contain [Evaluation Score]. Otherwise, we perform a re-plan to improve the previous action, and the response will include both [Evaluation Score] and [Improvement Feedback].

Below we will briefly show a few examples of the LLM's planning corrected by advantage feedback, using ReAd-S to complete the Sweep Floor task.

---

[Alice Response]:
Bob, we have green_cube_1 and green_cube_2 in the dustpan. Let's MOVE to the trash_bin so I can DUMP these cubes. After that, we can go for any remaining cubes.
PROCEED
NAME Alice ACTION MOVE trash_bin.
[Advantage Feedback]:
[Evaluation Score]:
The advantage score of Alice's action is -0.6652!
The advantage score is low and the plan need to be changed!
[Improve Feedback]
Alice's previous response:
Bob, we have green_cube_1 and green_cube_2 in the dustpan. Let's MOVE to the trash_bin so I can DUMP these cubes. After that, we can go for any remaining cubes.
PROCEED
NAME Alice ACTION MOVE trash_bin
The advantage score is -0.665, it is too low to run. The answer may contain a misunderstanding of the state of the environment, you must change your action to raise the advantage score over 0!
[Alice Response]:
Bob, I made a mistake in my previous plan. Since we already have green_cube_1 and green_cube_2 in the dustpan, I should DUMP them into the trash_bin. I'll do that while you WAIT.
PROCEED
NAME Alice ACTION DUMP
[Advantage Feedback]:
[Evaluation Score]:
The advantage score of Alice's action is 17.376!
The advantage score is high enough.

---

**Explain.** In this example, the MOVE action can only be used for cubes due to the task setting, and the MOVE of the trash bin is illegal. However, the LLMs learned from the web may consider the MOVE action can also be applied to the trash bin, which does not ground well with the specific task, resulting in the fact of hallucination. Our method can correct this type of error with advantage feedback according to the reward since taking MOVE action to the trash bin will obtain a zero reward and lead to the same next state as the current state.

---

[Alice Response]:
Bob, we have green_cube_1 and green_cube_2 in the dustpan. Let's MOVE to the next cube we need to sweep. I suggest we go to pink_cube_1 since it's the closest cube on the table. Once we're in position, I will WAIT for you to SWEEP it.
PROCEED
NAME Alice ACTION MOVE pink_cube_1
[Advantage Feedback]:
[Evaluation Score]:
The advantage score of Alice's action is -0.0927!
The advantage score is low and the plan needs to be changed!
[Improve Feedback]

---

Alice's previous response:
Bob, we have green_cube_1 and green_cube_2 in the dustpan. Let's MOVE to the next cube we need to sweep. I suggest we go to pink_cube_1 since it's the closest cube on the table. Once we're in position, I will WAIT for you to SWEEP it.
PROCEED
NAME Alice ACTION MOVE pink_cube_1 The advantage score is -0.0927, it is too low to run. The answer may contain a misunderstanding of the state of the environment, you must change your action to raise the advantage score over 0!
[Alice Response]:
Bob, I made a mistake in my previous suggestion. We should not move to pink_cube_1 as it's not one of our targets. Instead, let's DUMP the cubes we have in the dustpan into the trash_bin. I will do that while you WAIT.
PROCEED
NAME Alice ACTION DUMP
[Advantage Feedback]:
[Evaluation Score]:
The advantage score of Alice's action is 18.4184!
The advantage score is high enough.

**Explain.** In this example, the target colors to sweep are yellow and green. Since the environment also includes many cubes with other colors, LLM may plan to sweep cubes with the wrong colors initially. In such cases, the advantage feedback is helpful to correct the plan generated by the LLM.

[Bob Response]:
Alice, I will WAIT as you move into position for yellow_cube_3.
PROCEED
NAME Bob ACTION WAIT
[Advantage Feedback]:
[Evaluation Score]:
The advantage score of Bob's action is -3.2211!
The score is low and the plan needs to be changed!
[Improve Feedback]
Bob's previous response:
Alice, I will WAIT as you move into position for yellow_cube_3.
PROCEED
NAME Bob ACTION WAIT
The advantage score score is -3.221, it is too low to run, the answer may contain a misunderstanding of the state of the environment, you must change your action to raise the advantage score over -0.2!
[Bob Response]:
Alice, since you are moving to yellow_cube_3, I will also MOVE to yellow_cube_3 to be ready to SWEEP it once you are in position.
PROCEED
NAME Bob ACTION MOVE yellow_cube_3
[Advantage Feedback]:
[Evaluation Score]:
The advantage score of Bob's action is 0.6383!
The advantage score is high enough.

**Explain.** In this example, the LLM plans for Bob to move to yellow_cube_3 with Alice, it can sweep directly into the dustpan in the next step. However, during the first LLM planning, the LLM plans the WAIT action for Bob. If Bob performs this action at this time, Alice can only wait for Bob to move to yellow_cube_3. Thus, our method performs replanning based on the advantage feedback to reduces the interaction steps to the environment.

