# OpenReview forum: "Towards Efficient LLM Grounding for Embodied Multi-Agent Collaboration"
_ICLR.cc/2025/Conference — ICLR 2025 Conference Withdrawn Submission_

### Official Review · Reviewer_omMY · 2024-10-30

**Soundness:** 2
**Presentation:** 2
**Contribution:** 2
**Rating:** 5
**Confidence:** 4

**Summary:**

This work discusses grounding LLM's common sense reasoning capability in physical environments. Specifically, it explores methods to not only generate appropriate actions but also effectively increase interaction efficiency when using Multi-Agent LLM agents in embodied environments. Based on the theoretical background of multi-agent systems, the authors utilize a learned advantage function as immediate feedback for LLM agents. The LLM is prompted to generate actions with high advantage values, and if the generated action plan has low advantage values, it is designed to perform replanning autonomously. The superiority of this framework is demonstrated through experiments across multiple environments.

**Strengths:**

1. Grounding LLM's reasoning prowess in physical environments is a highly promising and crucial field that can help leverage continuously evolving LLMs for real-world problem solving.

2. Addressing interaction costs in embodied environments is a challenging yet practical problem that can resolve various factors, including safety issues and LLM inference costs.

3. The paper is well-organized and written clearly without confusion, making it easily accessible to readers.

**Weaknesses:**

1. The main concern is the novelty of the contribution. Applying MARL's theoretical aspects to embodied multi-agent collaboration appears quite straightforward, and seems independent from the authors' stated goal (as written in Line 77) of enhancing LLMs' reasoning capabilities.

2. While the introduction addresses interaction efficiency as a major issue in prior work, the data collection efficiency of the proposed learning method should also be discussed. This TRPO-style on-policy learning approach is generally known to be less data-efficient than off-policy RL algorithms (even in single-agent settings).

3. I am concerned if the comparison between baselines is fair. For instance, was the same amount of environmental interaction required to create ReAd's training dataset also provided to ReAct and Reflexion?

4. If data collection through interaction is possible, I think the current environmental tasks are relatively simple. For example, using sparse rewards (success/fail) attached to the end of collected dataset trajectories for retrieval-augmented task planning could enhance LLM reasoning more simply and efficiently than the proposed method. Please refer to paper [1]. Could the authors include [1] as a baseline in their experimental comparisons?

[1] LLM-Planner: Few-Shot Grounded Planning for Embodied Agents with Large Language Models, CVPR 2023

**Questions:**

1. In Line 38, I wonder about the author's mentions of "lack of task-specific knowledge" instead of domain-specific or environment-specific knowledge. LLMs possess commonsense actionable knowledge (i.e., knowledge to perform tasks) but have never learned the characteristics of the actual environment where this knowledge should be applied.

2. What is the size of dataset \mathcal{D}?

3. Rather than directly fine-tuning the LLM, it uses prompting to employ the LLM as an advantage function optimizer. How is the policy iteration implemented where (i) the LLM (behavior policy) is fine-tuned using the advantage function learned from dataset \mathcal{D}, and (ii) this fine-tuned policy becomes the behavior policy to collect new data? Does dataset collection by the behavior policy occur only once?

4. What is the neural network architecture of the advantage function?

5. Are environment observations converted to text before being passed to the LLM agent?

6. Does using the Advantage function (or Q-function) directly as a policy significantly decrease performance? I'm curious about how a conventional discrete action space RL approach would perform, where Q-values are calculated for all actions and the action with the highest Q-value is selected.

---

### Official Review · Reviewer_JEeL · 2024-11-02

**Soundness:** 3
**Presentation:** 3
**Contribution:** 2
**Rating:** 3
**Confidence:** 4

**Summary:**

This paper proposes a novel algorithm Reinforced Advantage feedback (ReAd), which leverages LLM for multiagent planning. The paper tests the algorithm on two domains, RoCoBench and Overcooked. The results show that ReAd outperformed existing LLM-based baselines.

**Strengths:**

The idea of incorporating some critic in the LLM planner is promising and can mitigate hallucination among the LLMs. It's an interesting and important problem for the AI/LLM community.

The paper is well written and easy to understand. The experiment is well done and the analysis is thorough and detailed.

I particularly appreciate the extensive discussions and analyses presented in the appendix, as they openly address the model's robustness and limitations. This level of detail not only strengthens the credibility of the research but also provides valuable insights for future work in this area.

Overall, I applaud the authors for this well-written work.

**Weaknesses:**

1. Model: The main weakness of the proposed algorithm is its limited generalizability. It seems like the critic NN is trained over hundreds of plans/trajectories in each domain. If this is the case, then this model becomes very domain-specific and would need a new critic for each new domain. That defeats the purpose of using LLM as a planner, which is meant to be fast and easily adaptable to different domains so you don't have to build any domain-specific model. I wonder if you can use LLM (maybe with few-shot prompting) as a critic?

2. Baseline choice: The paper only considered LLM baselines. Since the model is essentially doing RL, why not include RL baselines as well? If you are already training the critic to estimate some reward functions, then why do you need the LLM planner at all? Or at least you can try a version where you use RL for planning and LLM for communication among agents. There're also a lot of existing work on Overcooked AI. Does your model outperform those baselines?

3. Baseline implementation: If I recall correctly, LLM agents such as MindAgent and React don't have communication modules. I encourage the authors to include more details on how these baselines are implemented. If these baselines don't actually allow communication among agents in DV-RoCoBench, then it's not a fair comparison.

4. Evaluation: The paper would be much stronger if it would include a human experiment/evaluation, which was used in MindAgent and RoCo.

5. Presentation: It would make the results more readable if you can put ReAd-J and ReAd-S next to each other in Fig 3.

**Questions:**

1. The authors argued that multimodal VLMs such as GPT4o don't always parse the scene reliably. Have you tried query GPT4o with both the agent prompt (shown in Fig 1) and image input and use GPT4o as the planner instead of the image parser?

2. Have you thought about using another LLM as a critic?

3. How does ReAd-J compare to existing RL baselines on Overcooked?

4. It looks like none of the other LLM baselines succeeded in a single episode in Overcooked. This looks strange. Could you explain why they failed? Is it because they didn't complete the task within a fixed number of timesteps or they actually never completed the task? How was the timestep limit set? Does the results change if you increase the limit?

5. Did you try the models on the original RoCoBench? Why create a new dataset instead of using the original one, which has been tested/used by many existing studies?

6. Do other LLM baselines (MindAgent, React) have agent communication in your DV-RoCoBench experiment?

---

### Official Review · Reviewer_2eoE · 2024-11-03

**Soundness:** 3
**Presentation:** 3
**Contribution:** 3
**Rating:** 6
**Confidence:** 4

**Summary:**

The paper introduces a novel feedback mechanism, Reinforced Advantage feedback (ReAd), which combines the self-improvement abilities of large language models (LLMs) with advantage decomposition techniques from multi-agent reinforcement learning (MARL). This closed-loop feedback strategy with the introduce of a critic module enhances cooperation in multi-agent settings by training LLMs to learn a sequential advantage function using critic regression. Experiments on two multi-agent benchmarks indicate that ReAd outperforms existing feedback-based baselines and state-of-the-art methods, highlighting its effectiveness in promoting collaborative strategies among LLMs.

**Strengths:**

The paper presents an innovative approach by leveraging MARL theories to enhance collaborative behavior in LLM-based agents within multi-agent environments. The theoretical foundation is well-developed, allowing readers to gain a deep understanding of the approach. Furthermore, the experimental results and their detailed analysis contribute valuable insights to the research community.

**Weaknesses:**

- Since the paper takes insights from marl theories, including some marl methods as a baseline will further strengthen the robustness of the results

- Even with the _Difficult Variants_ of RoCoBench, the proposed method has achieved a nearly 100 percent success rate, so it seems the task may still be too easy or oversimplified.

- The experimental results showing a 0% success rate for all baselines on the overcooked-ai benchmark need further explanation. The original overcooked-ai paper reported RL baselines with non-zero performance—why aren’t they included here?

- As noted by the authors, there is a simplification of the visual perception and the generalizability of ReAd to more complex scenarios remains uncertain.

**Questions:**

Please see the Weaknesses.

---

### Official Review · Reviewer_AxcY · 2024-11-04

**Soundness:** 3
**Presentation:** 4
**Contribution:** 3
**Rating:** 6
**Confidence:** 3

**Summary:**

This paper introduces a new method for grounding LLMs for collaborative embodied multi-agent applications that utilizes a critic to score advantage values of actions based on the actions of partner agents. The critic provides advantage score for action predictions of the LLM to eliminate the need for direct physical interaction with the environment at inference time. They provide a detailed theoretical justification for the method and demonstrate superior grounding and coordination in the RoCo-Bench and Overcooked-AI environments compared to LLM-based baselines.

**Strengths:**

S1: The method introduces a method to eliminate the need for direct physical interaction with the environment, instead utilizing a pre-trained critic to approximate an advantage score and internally replan the action.

S2: The authors provide an extensive theoretical justification and guarantee for advantage decomposition in the multi-agent setting and empirically demonstrate the success of their method on two different benchmarks in multiple settings

S3: Their method is able to generate direct low-level actions without the need for intermediate high-level action plans which is a significant advantage over previous methods

S4: The method does not require access to exact probability of the sampled action from LLMs and can be used with closed-source LLMs

**Weaknesses:**

W1: Lack of discussion about generalization to unseen partners. In practical coordination scenarios, partner agents might not employ the same algorithms as the ego-agent. It is essential for coordination methods to be robust to unseen partners.

W2: Limited test coverage on Overcooked AI benchmark - it looks like Reflexion and React have 0 % task completion rate. Experiments should consider more time steps as it might turn out that although Reflexion or React take more time, they could have a higher success rate. Additionally, Overcooked also has a multiple delivery objective, as performance in the second delivery might be improved by better time management during the first delivery. Testing only one delivery is insufficient.

W3: The method utilizes a critic model that is trained in the environment. Whereas the baselines are all training-free approaches which might be an unfair comparison.

W4: ““Choosing w at the current state s signifies all agents take no actions, then the next state s′ = s and the agents receive shared reward r(s, w) = 0 since w bring no changes to the environment.” is a strong assumption as wait states can lead to negative rewards in the environment depending on the task configuration.

W5: The Overcooked-AI environment should be evaluated with ReAd-S to ensure the reliability of the method in the decentralized setting

**Questions:**

Q1. Can this approach work reliably in the Zero-shot coordination (unseen agents) setting?

Q2. Does the success rate on Overcooked increase with more time steps?

Q3: Is it possible to utilize a different LLM with a lower cost to analyze the ReAd-S of this method in the Overcooked-AI setting? Are LLMs other than GPT-4 able to perform low-level control in Overcooked where the action space involves only navigation and interaction?

Q4: Why isn't ReAd-S compared with other methods using the Llama-3.1-70B model?

---

### Note · Authors · 2024-12-06

I have read and agree with the venue's withdrawal policy on behalf of myself and my co-authors.